# OpenCon: Open-world Contrastive Learning

**Yiyou Sun**                                                                *sunyiyou@cs.wisc.edu*
**Yixuan Li**                                                                 *sharonli@cs.wisc.edu*
*University of Wisconsin-Madison*

**Reviewed on OpenReview:** *https://openreview.net/forum?id=2wWJxtpFer*

## Abstract

Machine learning models deployed in the wild naturally encounter unlabeled samples from both known and novel classes. Challenges arise in learning from both the labeled and unlabeled data, in an open-world semi-supervised manner. In this paper, we introduce a new learning framework, open-world contrastive learning (OpenCon). OpenCon tackles the challenges of learning compact representations for *both known and novel classes*, and facilitates novelty discovery along the way. We demonstrate the effectiveness of OpenCon on challenging benchmark datasets and establish competitive performance. On the ImageNet dataset, OpenCon significantly outperforms the current best method by 11.9% and 7.4% on novel and overall classification accuracy, respectively. Theoretically, OpenCon can be rigorously interpreted from an EM algorithm perspective—minimizing our contrastive loss partially maximizes the likelihood by clustering similar samples in the embedding space. The code is available at `https://github.com/deeplearning-wisc/opencon`.

## 1 Introduction

Modern machine learning methods have achieved remarkable success (Sun et al., 2017; Van den Oord et al., 2018; Chen et al., 2020a; Caron et al., 2020; He et al., 2020; Zheng et al., 2021; Wu et al., 2021; Cha et al., 2021; Cui et al., 2021; Jiang et al., 2021; Gao et al., 2021; Zhong et al., 2021a; Zhao & Han, 2021; Fini et al., 2021; Tsai et al., 2022; Zhang et al., 2022b; Wang et al., 2022). Noticeably, the vast majority of learning algorithms have been driven by the closed-world setting, where the classes are assumed stationary and unchanged. This assumption, however, rarely holds for models deployed in the wild. One important characteristic of open world is that the model will naturally encounter novel classes. Considering a realistic scenario, where a machine learning model for recognizing products in e-commerce may encounter brand-new products together with old products. Similarly, an autonomous driving model can run into novel objects on the road, in addition to known ones. Under the setting, the model should ideally learn to distinguish not only the known classes, but also the novel categories. This problem is proposed as open-world semi-supervised learning (Cao et al., 2022) or generalized category discovery (Vaze et al., 2022). Research efforts have only started very recently to address this important and realistic problem.

Formally, we are given a labeled training dataset $\mathcal{D}_l$ as well as an unlabeled dataset $\mathcal{D}_u$. The labeled dataset contains samples that belong to a set of known classes, while the unlabeled dataset has a mixture of samples from *both the known and novel classes*. In practice, such unlabeled in-the-wild data can be collected almost for free upon deploying a model in the open world, and thus is available in abundance. Under the setting, our goal is to learn distinguishable representations for both known and novel classes simultaneously. While this setting naturally suits many real-world applications, it also poses unique challenges due to: (a) the lack of clear separation between known vs. novel data in $\mathcal{D}_u$, and (b) the lack of supervision for data in novel classes. Traditional representation learning methods are not designed for this new setting. For example, supervised contrastive learning (SupCon) (Khosla et al., 2020) only assumes the labeled set $\mathcal{D}_l$, without considering the unlabeled data $\mathcal{D}_u$. Weakly supervised contrastive learning (Zheng et al., 2021) assumes the same classes in labeled and unlabeled data, hence remaining closed-world and less generalizable to novel

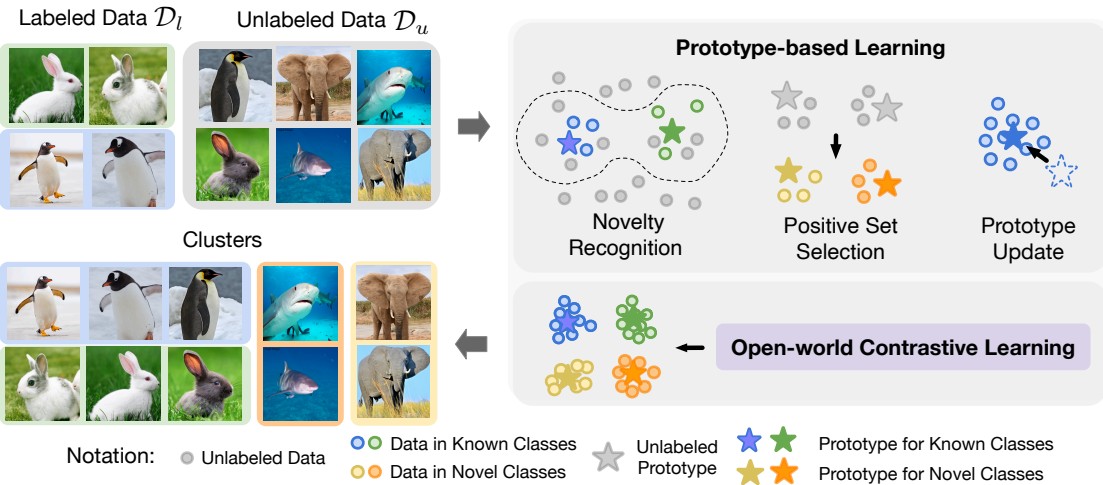

Figure 1: Illustration of our learning framework *Open-world Contrastive Learning* (OpenCon). The model is trained on a labeled dataset $\mathcal{D}_l$ of known classes, and an unlabeled dataset $\mathcal{D}_u$ (with samples from both known and novel classes). OpenCon aims to learn distinguishable representations for both known (blue and green) and novel (yellow and orange) classes simultaneously. See Section 3 for details.

samples. Self-supervised learning (Chen et al., 2020a) relies completely on the unlabeled set $\mathcal{D}_u$ and does not utilize the availability of the labeled dataset $\mathcal{D}_l$.

Targeting these challenges, we formally introduce a new learning framework, *open-world contrastive learning* (dubbed **OpenCon**). OpenCon is designed to produce a compact representation space for both known and novel classes, and facilitates novelty discovery along the way. Key to our framework, we propose a novel prototype-based learning strategy, which encapsulates two components. First, we leverage the prototype vectors to separate known vs. novel classes in unlabeled data $\mathcal{D}_u$. The prototypes can be viewed as a set of representative embeddings, one for each class, and are updated by the evolving representations. Second, to mitigate the challenge of lack of supervision, we generate pseudo-positive pairs for contrastive comparison. We define the positive set to be those examples carrying the same approximated label, which is predicted based on the closest class prototype. In effect, the loss encourages closely aligned representations to all samples from the same predicted class, rendering a compact clustering of the representation.

Our framework offers several compelling advantages. **(1)** Empirically, OpenCon establishes strong performance on challenging benchmark datasets, outperforming existing baselines by a significant margin (Section 5). OpenCon is also competitive without knowing the number of novel classes in advance—achieving similar or even slightly better performance compared to the oracle (in which the number of classes is given). **(2)** Theoretically, we demonstrate that our prototype-based learning can be rigorously interpreted from an Expectation-Maximization (EM) algorithm perspective. **(3)** Our framework is end-to-end trainable, and is compatible with both CNN-based and Transformer-based architectures. Our **main contributions** are:

1. We propose a novel framework, open-world contrastive learning (OpenCon), tackling a largely unexplored problem in representation learning. As an integral part of our framework, we also introduce a prototype-based learning algorithm, which facilitates novelty discovery and learning distinguishable representations.

2. Empirically, OpenCon establishes competitive performance on challenging tasks. For example, on the ImageNet dataset, OpenCon substantially outperforms the current best method ORCA (Cao et al., 2022) by **11.9**% and **7.4**% in terms of novel and overall accuracy.

3. We provide insights through extensive ablations, showing the effectiveness of components in our framework. Theoretically, we show a formal connection with EM algorithm—minimizing our contrastive loss partially maximizes the likelihood by clustering similar samples in the embedding space.

Table 1: Comparison of different problem settings.

| Problem Setting | Labeled data | Unlabeled data | |
| --- | --- | --- | --- |
| | | Known classes | Novel classes |
| Semi-supervised learning | Yes | Yes | No |
| Robust semi-supervised learning | Yes | Yes | Yes (Reject) |
| Supervised learning | Yes | No | No |
| Novel class discovery | Yes | No | Yes (Discover) |
| Open-world Semi-supervised learning | Yes | Yes | Yes (Cluster) |

## 2    Problem Setup

The open-world setting emphasizes the fact that new classes can emerge jointly with existing classes in the wild. Formally, we describe the data setup and learning goal:

**Data setup** We consider the training dataset $\mathcal{D} = \mathcal{D}_l \cup \mathcal{D}_u$ with two parts:

1. The labeled set $\mathcal{D}_l = \{\mathbf{x}_i, y_i\}_{i=1}^n$, with $y_i \in \mathcal{Y}_l$. The label set $\mathcal{Y}_l$ is known.

2. The unlabeled set $\mathcal{D}_u = \{\mathbf{x}_i\}_{i=1}^m$, where each sample $\mathbf{x}_i \in \mathcal{X}$ can come from either known or novel classes[1]. Note that we do not have access to the labels in $\mathcal{D}_u$. For mathematical convenience, we denote the underlying label set as $\mathcal{Y}_{\text{all}}$, where $\mathcal{Y}_l \subset \mathcal{Y}_{\text{all}}$ implies category shift and expansion. Accordingly, the set of novel classes is $\mathcal{Y}_n = \mathcal{Y}_{\text{all}} \backslash \mathcal{Y}_l$, where the *subscript n* stands for "**n**ovel". The model has no knowledge of the set $\mathcal{Y}_n$ nor its size.

**Goal** Under the setting, the goal is to learn distinguishable representations *for both known and novel classes* simultaneously.

**Difference w.r.t. existing problem settings** Our paper addresses a practical and underexplored problem, which differs from existing problem settings (see Table 1 for a summary). In particular, (a) we consider *both labeled data and unlabeled data* in training, and (b) we consider a mixture of *both known and novel classes* in unlabeled data. Note that our setting generalizes traditional representation learning. For example, Supervised Contrastive Learning (SupCon) (Khosla et al., 2020) only assumes the labeled set $\mathcal{D}_l$, without considering the unlabeled data $\mathcal{D}_u$. Weakly supervised contrastive learning (Zheng et al., 2021) assumes the same classes in labeled and unlabeled data, *i.e.*, $\mathcal{Y}_l = \mathcal{Y}_{\text{all}}$, and hence remains closed-world. Self-supervised learning (Chen et al., 2020a) relies completely on the unlabeled set $\mathcal{D}_u$ and does not assume the availability of the labeled dataset. The setup is also known as open-world semi-supervised learning, which is originally introduced by (Cao et al., 2022). Despite the similar setup, our learning goal is fundamentally different: Cao et al. (2022) focus on classification accuracy, but not learning high-quality embeddings. Similar to GCD Vaze et al. (2022), we aim to learning compact embeddings for both known and novel classes.

## 3    Proposed Method: Open-world Contrastive Learning

We formally introduce a new learning framework, *open-world contrastive learning* (dubbed OpenCon), which is designed to produce compact representation space for both known and novel classes. The open-world setting posits unique challenges for learning effective representations, namely due to (1) the lack of the separation between known vs. novel data in $\mathcal{D}_u$, (2) the lack of supervision for data in novel classes. Our learning framework targets these challenges.

### 3.1    Background: Generalized Contrastive Loss

We start by defining a generalized contrastive loss that can characterize the family of contrastive losses. We will later instantiate the formula to define our open-world contrastive loss (Section 3.2 and Section 3.3). Specifically, we consider a deep neural network encoder $\phi : \mathcal{X} \mapsto \mathbb{R}^d$ that maps the input $\mathbf{x}$ to a $L_2$-normalized

---

[1]This is different from the problem of Novel Class Discovery, which assumes the unlabeled set is purely from novel classes.

feature embedding $\phi(\mathbf{x})$. Contrastive losses operate on the normalized feature $\mathbf{z} = \phi(\mathbf{x})$. In other words, the features have unit norm and lie on the unit hypersphere. For a given anchor point $\mathbf{x}$, we define the per-sample contrastive loss:

$$\mathcal{L}_\phi\big(\mathbf{x}; \tau, \mathcal{P}(\mathbf{x}), \mathcal{N}(\mathbf{x})\big) = -\frac{1}{|\mathcal{P}(\mathbf{x})|} \sum_{\mathbf{z}^+ \in \mathcal{P}(\mathbf{x})} \log \frac{\exp(\mathbf{z}^\top \cdot \mathbf{z}^+/\tau)}{\sum_{\mathbf{z}^- \in \mathcal{N}(\mathbf{x})} \exp(\mathbf{z}^\top \cdot \mathbf{z}^-/\tau)}, \tag{1}$$

where $\tau$ is the temperature parameter, $\mathbf{z}$ is the $L_2$-normalized embedding vector of $\mathbf{x}$, $\mathcal{P}(\mathbf{x})$ is the positive set of embeddings *w.r.t.* $\mathbf{z}$, and $\mathcal{N}(\mathbf{x})$ is the negative set of embeddings.

In open-world contrastive learning, the crucial challenge is how to construct $\mathcal{P}(\mathbf{x})$ and $\mathcal{N}(\mathbf{x})$ *for different types of samples*. Recall that we have two broad categories of training data: (1) labeled data $\mathcal{D}_l$ with known class, and (2) unlabeled data $\mathcal{D}_u$ with both known and novel classes. In conventional supervised CL frameworks with $\mathcal{D}_l$ only, the positive sample pairs can be easily drawn according to the ground-truth labels (Khosla et al., 2020). That is, $\mathcal{P}(\mathbf{x})$ consists of embeddings of samples that carry the same label as the anchor point $\mathbf{x}$, and $\mathcal{N}(\mathbf{x})$ contains all the embeddings in the multi-viewed mini-batch excluding itself. However, this is not straightforward in the open-world setting with novel classes.

### 3.2 Learning from Wild Unlabeled Data

We now dive into the most challenging part of the data, $\mathcal{D}_u$, which contains both known and novel classes. We propose a novel prototype-based learning strategy that tackles the challenges of: (1) the separation between known and novel classes in $\mathcal{D}_u$, and (2) pseudo label assignment that can be used for positive set construction for novel classes. Both components facilitate the goal of learning compact representations, and enable end-to-end training.

Key to our framework, we keep a prototype embedding vector $\boldsymbol{\mu}_c$ for each class $c \in \mathcal{Y}_{\text{all}}$. Here $\mathcal{Y}_{\text{all}}$ contains both known classes $\mathcal{Y}_l$ and novel classes $\mathcal{Y}_n = \mathcal{Y}_{\text{all}} \backslash \mathcal{Y}_l$, and $\mathcal{Y}_l \cap \mathcal{Y}_n = \emptyset$. The prototypes can be viewed as a set of representative embedding vectors. All the prototype vectors $\mathbf{M} = [\boldsymbol{\mu}_1 | ... | \boldsymbol{\mu}_c | ...]_{c \in \mathcal{Y}_{\text{all}}}$ are randomly initiated at the beginning of training, and will be updated along with learned embeddings. We will also discuss determining the cardinality $|\mathcal{Y}_{\text{all}}|$ (*i.e.*, number of prototypes) in Section 6.

**Prototype-based OOD detection** We leverage the prototype vectors to perform out-of-distribution (OOD) detection, *i.e.*, separate known vs. novel data in $\mathcal{D}_u$. For any given sample $\mathbf{x}_i \in \mathcal{D}_u$, we measure the cosine similarity between its embedding $\phi(\mathbf{x}_i)$ and prototype vectors of known classes $\mathcal{Y}_l$. If the sample embedding is far away from all the known class prototypes, it is more likely to be a novel sample, and vice versa. Formally, we propose the level set estimation:

$$\mathcal{D}_n = \{\mathbf{x}_i | \max_{j \in \mathcal{Y}_l} \quad \boldsymbol{\mu}_j^\top \cdot \phi(\mathbf{x}_i) < \lambda\}, \tag{2}$$

where a thresholding mechanism is exercised to distinguish between known and novel samples during training time. The threshold $\lambda$ can be chosen based on the labeled data $\mathcal{D}_l$. Specifically, one can calculate the scores $\max_{j \in \mathcal{Y}_l} \boldsymbol{\mu}_j^\top \cdot \phi(\mathbf{x}_i)$ for all the samples in $\mathcal{D}_l$, and use the score at the $p$-percentile as the threshold. For example, when $p = 90$, that means 90% of labeled data is above the threshold. We provide ablation on the effect of $p$ later in Section 6 and theoretical insights into why OOD detection helps open-world representation learning in Appendix D.

**Positive and negative set selection** Now that we have identified novel samples from the unlabeled sample, we would like to facilitate learning compact representations for $\mathcal{D}_n$, where samples belonging to the same class are close to each other. As mentioned earlier, the crucial challenge is how to construct the positive set, denoted as $\mathcal{P}_n(\mathbf{x})$. In particular, we do not have any supervision signal for unlabeled data in the novel classes. We propose utilizing the predicted label $\hat{y} = \text{argmax}_{j \in \mathcal{Y}_{\text{all}}} \boldsymbol{\mu}_j^\top \cdot \phi(\mathbf{x})$ for positive set selection.

For a mini-batch $\mathcal{B}_n$ with samples drawn from $\mathcal{D}_n$, we apply two random augmentations for each sample and generate a multi-viewed batch $\tilde{\mathcal{B}}_n$. We denote the embeddings of the multi-viewed batch as $\mathcal{A}_n$, where the cardinality $|\mathcal{A}_n| = 2|\mathcal{B}_n|$. For any sample $\mathbf{x}$ in the mini-batch $\tilde{\mathcal{B}}_n$, we propose selecting the positive and

negative set of embeddings as follows:

$$\mathcal{P}_n(\mathbf{x}) = \{\mathbf{z}'|\mathbf{z}' \in \{\mathcal{A}_n \backslash \mathbf{z}\}, \hat{y}' = \hat{y}\}, \tag{3}$$
$$\mathcal{N}_n(\mathbf{x}) = \mathcal{A}_n \backslash \mathbf{z}, \tag{4}$$

where $\mathbf{z}$ is the $L_2$-normalized embedding of $\mathbf{x}$, and $\hat{y}'$ is the predicted label for the corresponding training example of $\mathbf{z}'$. In other words, we define the positive set of $\mathbf{x}$ to be those examples carrying the same *approximated* label prediction $\hat{y}$.

With the positive and negative sets defined, we are now ready to introduce our new contrastive loss for open-world data. We desire embeddings where samples assigned with the same pseudo-label can form a compact cluster. Following the general template in Equation 1, we define a novel loss function:

$$\mathcal{L}_n = \sum_{\mathbf{x} \in \tilde{\mathcal{B}}_n} \mathcal{L}_\phi\big(\mathbf{x}; \tau_n, \mathcal{P}_n(\mathbf{x}), \mathcal{N}_n(\mathbf{x})\big). \tag{5}$$

For each anchor, the loss encourages the network to align embeddings of its positive pairs while repelling the negatives. *All* positives in a multi-viewed batch (*i.e.*, the augmentation-based sample as well as any of the remaining samples with the same label) contribute to the numerator. The loss encourages the encoder to give closely aligned representations to *all* entries from the same *predicted* class, resulting in a compact representation space. We provide visualization in Figure 2 (right).

**Prototype update**  The most canonical way to update the prototype embeddings is to compute it in every iteration of training. However, this would extract a heavy computational toll and in turn cause unbearable training latency. Instead, we update the class-conditional prototype vector in a moving-average style (Li et al., 2020; Wang et al., 2022):

$$\boldsymbol{\mu}_c := \text{Normalize}\left(\gamma \boldsymbol{\mu}_c + (1 - \gamma)\mathbf{z}\right), \text{for } c = \begin{cases} y \text{ (ground truth label)}, & \text{if } \mathbf{z} \in \mathcal{D}_l \\ \text{argmax}_{j \in \mathcal{Y}_n} \boldsymbol{\mu}_j^\top \cdot \mathbf{z}, & \text{if } \mathbf{z} \in \mathcal{D}_n \end{cases} \tag{6}$$

Here, the prototype $\boldsymbol{\mu}_c$ of class $c$ is defined by the moving average of the normalized embeddings $\mathbf{z}$, whose predicted class conforms to $c$. $\mathbf{z}$ are embeddings of samples from $\mathcal{D}_l \cup \mathcal{D}_n$. $\gamma$ is a tunable hyperparameter.

**Remark:** We exclude samples in $\mathcal{D}_u \backslash \mathcal{D}_n$ because they may contain non-distinguishable data from known and unknown classes, which undesirably introduce noise to the prototype estimation. We verify this phenomenon by comparing the performance of mixing $\mathcal{D}_u \backslash \mathcal{D}_n$ with labeled data $\mathcal{D}_l$ for training the known classes. The results verify our hypothesis that the non-distinguishable data would be harmful to the overall accuracy. We provide more discussion on this in Appendix E.

### 3.3 Open-world Contrastive Loss

Putting it all together, we define the open-world contrastive loss (dubbed **OpenCon**) as the following:

$$\mathcal{L}_{\text{OpenCon}} = \lambda_n \mathcal{L}_n + \lambda_l \mathcal{L}_l + \lambda_u \mathcal{L}_u, \tag{7}$$

where $\mathcal{L}_n$ is the newly devised contrastive loss for the novel data, $\mathcal{L}_l$ is the supervised contrastive loss (Khosla et al., 2020) employed on the labeled data $\mathcal{D}_l$, and $\mathcal{L}_u$ is the self-supervised contrastive loss (Chen et al., 2020a) employed on the unlabeled data $\mathcal{D}_u$. $\lambda$ are the coefficients of loss terms. Details of $\mathcal{L}_l$ and $\mathcal{L}_u$ are in Appendix B, along with the complete pseudo-code in Algorithm 1 (Appendix).

**Remark**  Our loss components work collaboratively to enhance the embedding quality in an open-world setting. The overall objective well suits the complex nature of our training data, which blends both labeled and unlabeled data. As we will show later in Section 6, a simple solution by combining supervised contrastive loss (on labeled data) and self-supervised loss (on unlabeled data) is suboptimal. Instead, having $\mathcal{L}_n$ is critical to encourage closely aligned representations to *all* entries from the same predicted class, resulting in an overall more compact representation for novel classes.

# 4   Theoretical Understandings

**Overview**   Our learning objective using wild data (*c.f.* Section 3.2) can be rigorously interpreted from an Expectation-Maximization (EM) algorithm perspective. We start by introducing the high-level ideas of how our method can be decomposed into E-step and M-step respectively. At the **E-step**, we assign each data example $\mathbf{x} \in \mathcal{D}_n$ to one specific cluster. In OpenCon, it is estimated by using the prototypes: $\hat{y}_i = \arg\max_{j \in \mathcal{Y}_{\text{all}}} \boldsymbol{\mu}_j^\top \cdot \phi(\mathbf{x}_i)$. At the **M-step**, the EM algorithm aims to maximize the likelihood under the posterior class probability from the previous E-step. Theoretically, we show that minimizing our contrastive loss $\mathcal{L}_n$ (Equation 5) partially maximizes the likelihood by clustering similar examples. In effect, our loss concentrates similar data to the corresponding prototypes, encouraging the compactness of features.

## 4.1   Analyzing the E-step

In **E-step**, the goal of the EM algorithm is to maximize the likelihood with learnable feature encoder $\phi$ and prototype matrix $\mathbf{M} = [\boldsymbol{\mu}_1|...|\boldsymbol{\mu}_c|...]$, which can be lower bounded:

$$\sum_i^{|\mathcal{D}_n|} \log p(\mathbf{x}_i|\phi, \mathbf{M}) \geq \sum_i^{|\mathcal{D}_n|} q_i(c) \log \sum_{c \in \mathcal{Y}_{\text{all}}} \frac{p(\mathbf{x}_i, c|\phi, \mathbf{M})}{q_i(c)},$$

where $q_i(c)$ is denoted as the density function of a possible distribution over $c$ for sample $\mathbf{x}_i$. By using the fact that $\log(\cdot)$ function is concave, the inequality holds with equality when $\frac{p(\mathbf{x}_i, c|\phi, \mathbf{M})}{q_i(c)}$ is a constant value, therefore we set:

$$q_i(c) = \frac{p(\mathbf{x}_i, c|\phi, \mathbf{M})}{\sum_{c \in \mathcal{Y}_{\text{all}}} p(\mathbf{x}_i, c|\phi, \mathbf{M})} = \frac{p(\mathbf{x}_i, c|\phi, \mathbf{M})}{p(\mathbf{x}_i|\phi, \mathbf{M})} = p(c|\mathbf{x}_i, \phi, \mathbf{M}),$$

which is the posterior class probability. To estimate $p(c|\mathbf{x}_i, \phi, \mathbf{M})$, we model the data using the von Mises-Fisher (vMF) (Fisher, 1953) distribution since the normalized embedding locates in a high-dimensional hyperspherical space.

**Assumption 4.1.** The density function is given by $f(\mathbf{x}|\boldsymbol{\mu}, \kappa) = c_d(\kappa)e^{\kappa \boldsymbol{\mu}^\top \phi(\mathbf{x})}$, where $\kappa$ is the concentration parameter and $c_d(\kappa)$ is a coefficient.

With the vMF distribution assumption in 4.1, we have $p(c|\mathbf{x}_i, \phi, \mathbf{M}) = \sigma_c(\mathbf{M}^\top \cdot \phi(\mathbf{x}_i))$, where $\sigma$ denotes the softmax function and $\sigma_c$ is the $c$-th element. Empirically we take a one-hot prediction with $\hat{y}_i = \arg\max_{j \in \mathcal{Y}_{\text{all}}} \boldsymbol{\mu}_j^\top \cdot \phi(\mathbf{x}_i)$ since each example inherently belongs to exactly one prototype, so we let $q_i(c) = \mathbf{1}\{c = \hat{y}_i\}$.

## 4.2   Analyzing the M-step

In **M-step**, using the label distribution prediction $q_i(c)$ in the E-step, the optimization for the network $\phi$ and the prototype matrix $\mathbf{M}$ is given by:

$$\arg\max_{\phi, \mathbf{M}} \sum_{i=1}^{|\mathcal{D}_n|} \sum_{c \in \mathcal{Y}_{\text{all}}} q_i(c) \log \frac{p(\mathbf{x}_i, c|\phi, \mathbf{M})}{q_i(c)} \tag{8}$$

The joint optimization target in Equation 8 is then achieved by rewriting the Equation 8 according to the following Lemma 4.2 with proof in Appendix F:

**Lemma 4.2.** *(Zha et al., 2001) We define the set of samples with the same prediction $\mathcal{S}(c) = \{\mathbf{x}_i \in \mathcal{D}_n | \hat{y}_i = c\}$. The maximization step is equivalent to aligning the feature vector $\phi(\mathbf{x})$ to the corresponding prototype $\boldsymbol{\mu}_c$:*

$$\arg\max_{\phi, \mathbf{M}} \sum_{i=1}^{|\mathcal{D}_n|} \sum_{c \in \mathcal{Y}_{all}} q_i(c) \log \frac{p(\mathbf{x}_i, c|\phi, \mathbf{M})}{q_i(c)} = \arg\max_{\phi, \mathbf{M}} \sum_{c \in \mathcal{Y}_{all}} \sum_{\mathbf{x} \in \mathcal{S}(c)} \phi(\mathbf{x})^\top \cdot \boldsymbol{\mu}_c$$

In our algorithm, the maximization step is achieved by optimizing $\mathbf{M}$ and $\phi$ separately.

(a) **Optimizing M**:

For fixed $\phi$, the optimal prototype is given by $\boldsymbol{\mu}_c^* = \text{Normalize}(\mathbb{E}_{\mathbf{x} \in \mathcal{S}(c)}[\phi(\mathbf{x})])$. **This optimal form empirically corresponds to our prototype estimation in Equation 6.** Empirically, it is expensive to collect all features in $\mathcal{S}(c)$. We use the estimation of $\boldsymbol{\mu}_c$ by moving average: $\boldsymbol{\mu}_c := \text{Normalize}\left(\gamma \boldsymbol{\mu}_c + (1 - \gamma)\phi(\mathbf{x})\right), \forall \mathbf{x} \in \mathcal{S}(c)$.

(b) **Optimizing $\phi$**:

We then show that the contrastive loss $\mathcal{L}_n$ composed with the alignment loss part $\mathcal{L}_a$ encourages the closeness of features from positive pairs. By minimizing $\mathcal{L}_a$, it is approximately maximizing the target in Equation 8 with the optimal prototypes $\boldsymbol{\mu}_c^*$. We can decompose the loss as follows:

$$
\begin{aligned}
\mathcal{L}_n &= -\frac{1}{|\mathcal{P}(\mathbf{x})|} \sum_{\mathbf{z}^+ \in \mathcal{P}(\mathbf{x})} \log \frac{\exp(\mathbf{z}^\top \cdot \mathbf{z}^+ / \tau)}{\sum_{\mathbf{z}^- \in \mathcal{N}(\mathbf{x})} \exp(\mathbf{z} \cdot \mathbf{z}^- / \tau)} \\
&= \underbrace{-\frac{1}{|\mathcal{P}(\mathbf{x})|} \sum_{\mathbf{z}^+ \in \mathcal{P}(\mathbf{x})} (\mathbf{z}^\top \cdot \mathbf{z}^+ / \tau)}_{\mathcal{L}_a(\mathbf{x})} + \underbrace{\frac{1}{|\mathcal{P}(\mathbf{x})|} \sum_{\mathbf{z}^+ \in \mathcal{P}(\mathbf{x})} \log \sum_{\mathbf{z}^- \in \mathcal{N}(\mathbf{x})} \exp(\mathbf{z}^\top \cdot \mathbf{z}^- / \tau)}_{\mathcal{L}_b(\mathbf{x})}.
\end{aligned}
$$

In particular, the first term $\mathcal{L}_a(\mathbf{x})$ is referred to as the alignment term (Wang & Isola, 2020), which encourages the compactness of features from positive pairs. To see this, we have the following lemma 4.3 with proof in Appendix F.

**Lemma 4.3.** *Minimizing $\mathcal{L}_a(\mathbf{x})$ is equivalent to the maximization step w.r.t. parameter $\phi$.*

$$
\underset{\phi}{\arg\min} \sum_{\mathbf{x} \in \mathcal{D}_n} \mathcal{L}_a(\mathbf{x}) = \underset{\phi}{\arg\max} \sum_{c \in \mathcal{Y}_{all}} \sum_{\mathbf{x} \in \mathcal{S}(c)} \phi(\mathbf{x})^\top \cdot \boldsymbol{\mu}_c^*,
$$

**Summary** These observations validate that our framework learns representation for novel classes in an EM fashion. Importantly, we extend EM from a traditional learning setting to an open-world setting with the capability to handle real-world data arising in the wild. We proceed by introducing the empirical verification of our algorithm.

## 5 Experimental Results

**Datasets** We evaluate on standard benchmark image classification datasets CIFAR-100 (Krizhevsky et al., 2009) and ImageNet (Deng et al., 2009). For the ImageNet, we sub-sample 100 classes, following the same setting as ORCA (Cao et al., 2022) for fair comparison. Results on ImageNet-1k is also provided in Appendix L. Note that we focus on these tasks, as they are much more challenging than toy datasets with fewer classes. The additional comparison on CIFAR-10 is in Appendix G. By default, classes are divided into 50% seen and 50% novel classes. We then select 50% of known classes as the labeled dataset, and the rest as the unlabeled set. The division is consistent with Cao et al. (2022), which allows us to compare the performance in a fair setting. Additionally, we explore different ratios of unlabeled data and novel classes (see Section 6).

**Evaluation metrics** We follow the evaluation strategy in Cao et al. (2022) and report the following metrics: (1) classification accuracy on known classes, (2) classification accuracy on the novel data, and (3) overall accuracy on all classes. The accuracy of the novel classes is measured by solving an optimal assignment problem using the Hungarian algorithm (Kuhn & Yaw, 1955). When reporting accuracy on all classes, we solve optimal assignments using both known and novel classes.

**Experimental details** We use ResNet-18 as the backbone for CIFAR-100 and ResNet-50 as the backbone for ImageNet-100. The pre-trained backbones (no final FC layer) are identical to the ones in Cao et al. (2022). To ensure a fair comparison, we follow the same practice in Cao et al. (2022) and only update the

Table 2: Main Results. Asterisk ($\star$) denotes that the original method can not recognize seen classes. Dagger ($\dagger$) denotes the original method can not detect novel classes (and we had to extend it). Results on ORCA, GCD and OpenCon (mean and standard deviation) are averaged over five different runs. The ORCA results are reported based on the official repo (Cao et al.).

| Method | CIFAR-100 | | | ImagNet-100 | | |
|---|---|---|---|---|---|---|
| | All | Novel | Seen | All | Novel | Seen |
| $\dagger$**FixMatch** (Kurakin et al., 2020) | 20.3 | 23.5 | 39.6 | 34.9 | 36.7 | 65.8 |
| $\dagger$**DS$^3$L** (Guo et al., 2020) | 24.0 | 23.7 | 55.1 | 30.8 | 32.5 | 71.2 |
| $\dagger$**CGDL** (Sun et al., 2020) | 23.6 | 22.5 | 49.3 | 31.9 | 33.8 | 67.3 |
| $\star$**DTC** (Han et al., 2019) | 18.3 | 22.9 | 31.3 | 21.3 | 20.8 | 25.6 |
| $\star$**RankStats** (Zhao & Han, 2021) | 23.1 | 28.4 | 36.4 | 40.3 | 28.7 | 47.3 |
| $\star$**SimCLR** (Chen et al., 2020a) | 22.3 | 21.2 | 28.6 | 36.9 | 35.7 | 39.5 |
| **ORCA** (Cao et al., 2022) | $47.2^{\pm0.7}$ | $41.0^{\pm1.0}$ | $66.7^{\pm0.2}$ | $76.4^{\pm1.3}$ | $68.9^{\pm0.8}$ | $89.1^{\pm0.1}$ |
| **GCD** (Vaze et al., 2022) | $46.8^{\pm0.5}$ | $43.4^{\pm0.7}$ | $\mathbf{69.7}^{\pm0.4}$ | $75.5^{\pm1.4}$ | $72.8^{\pm1.2}$ | $\mathbf{90.9}^{\pm0.2}$ |
| **OpenCon (Ours)** | $\mathbf{52.7}^{\pm0.6}$ | $\mathbf{47.8}^{\pm0.6}$ | $69.1^{\pm0.3}$ | $\mathbf{83.8}^{\pm0.3}$ | $\mathbf{80.8}^{\pm0.3}$ | $90.6^{\pm0.1}$ |

parameters of the last block of ResNet. In addition, we add a trainable two-layer MLP projection head that projects the feature from the penultimate layer to a lower-dimensional space $\mathbb{R}^d$ ($d = 128$), which is shown to be effective for contrastive loss (Chen et al., 2020a). We use the same data augmentation strategies as SimCLR (Chen et al., 2020a). Same as in Cao et al. (2022), we regularize the KL-divergence between the predicted label distribution $p(\hat{y})$ and the class prior to prevent the network degenerating into a trivial solution in which all instances are assigned to a few classes. We provide extensive details on the training configurations and all hyper-parameters in Appendix I.

**OpenCon achieves SOTA performance** As shown in Table 1, OpenCon outperforms the rivals by a significant margin on both CIFAR and ImageNet datasets. Our comparison covers an extensive collection of algorithms, including the best-performed methods to date. In particular, on ImageNet-100, we improve upon the best baseline by **7.4**% in terms of overall accuracy. It is also worth noting that OpenCon improves the accuracy of novel classes by **11.9**%. Note that the open-world representation learning is a relatively new setting. Closest to our setting is the open-world semi-supervised learning (SSL) algorithms, namely ORCA (Cao et al., 2022) and GCD (Vaze et al., 2022)—that directly optimize the classification performance. While our framework emphasizes representation learning, we demonstrate the quality of learned embeddings by also measuring the classification accuracy. This can be easily done by leveraging our learned prototypes on a converged model: $\hat{y} = \operatorname{argmax}_{j \in \mathcal{Y}_{\text{all}}} \boldsymbol{\mu}_j^\top \cdot \phi(\mathbf{x})$. We discuss the significance *w.r.t.* existing works in detail:

- **OpenCon vs. ORCA** Our framework bears significant differences *w.r.t.* ORCA in terms of learning goal and approach. (1) Our framework focuses on the representation learning problem, whereas ORCA optimizes for the classification performance using cross-entropy loss. Unlike ours, ORCA does not necessarily learn compact representations, as evidenced in Figure 2 (left). (2) We propose a novel open-world contrastive learning framework, whereas ORCA does not employ contrastive learning. ORCA uses a pairwise loss to predict similarities between pairs of instances, and does not consider negative samples. In contrast, our approach constructs both positive and negative sample sets, which encourage aligning representations to *all* entries from the same ground-truth label or predicted pseudo label (for novel classes). (3) Our framework explicitly considers OOD detection, which allows separating known vs. novel data in $\mathcal{D}_u$. ORCA does not consider this and can suffer from noise in the pairwise loss (*e.g.*, the loss may maximize the similarity between samples from known vs. novel classes).

- **OpenCon vs. GCD** There are two key differences to highlight: (1) GCD (Vaze et al., 2022) requires a two-stage training procedure, whereas our learning framework proposes an end-to-end training strategy. Specifically, GCD applies the SupCon loss (Khosla et al., 2020) on the labeled data $\mathcal{D}_l$ and SimCLR loss (Chen et al., 2020a) on the unlabeled data $\mathcal{D}_u$. The feature is then clustered separately by a semi-supervised K-means method. However, the two-stage method hinders the useful pseudo-labels to be incorporated into the training stage, which results in suboptimal performance. In contrast, our

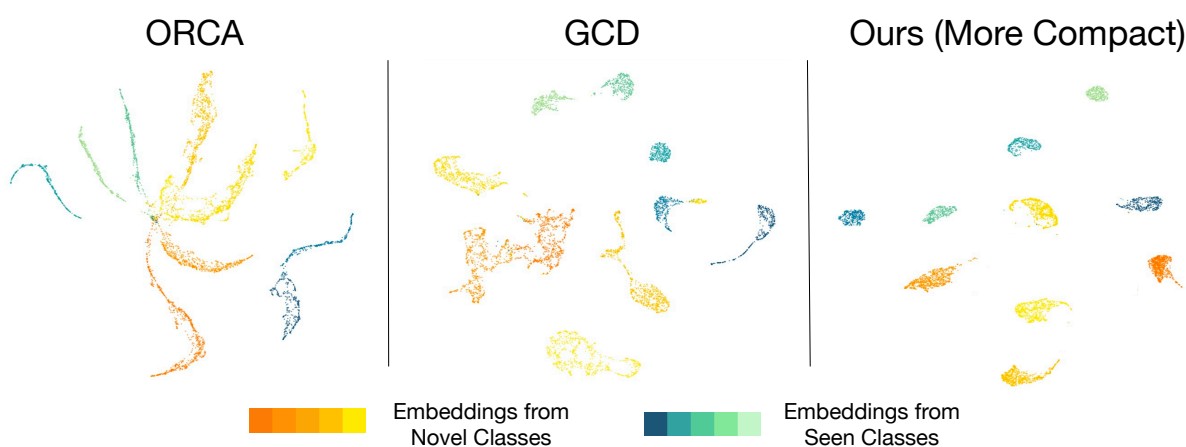

Figure 2: UMAP (McInnes et al., 2018) visualization of the feature embedding from 10 classes (5 for seen, 5 for novel) when the model is trained on ImageNet-100 with ORCA (Cao et al., 2022), GCD (Vaze et al., 2022) and OpenCon (ours).

prototype-based learning strategy alleviates the need for a separate clustering process (*c.f.* Section 3.2), which is therefore easy to use in practice and provides meaningful supervision for the unlabeled data. (2) We propose a contrastive loss $\mathcal{L}_n$ better utilizing the pseudo-labels during training, which facilitates learning a more compact representation space for the novel data . From Table 2, we observe that OpenCon outperforms GCD by **8.3**% (overall accuracy) on ImageNet-100, showcasing the benefits of our framework.

Lastly, for completeness, we compare methods in related problem domains: (1) *novel class detection*: DTC (Han et al., 2019), RANKSTATS (Zhao & Han, 2021), (2) *semi-supervised learning*: FIXMATCH (Kurakin et al., 2020), DS³L (Guo et al., 2020) and CGDL (Sun et al., 2020). We also compare it with the common representation method SIMCLR (Chen et al., 2020a). These methods are not designed for the Open-SSL task, therefore the performance is less competitive.

**OpenCon is competitive on ViT** Going beyond convolutional neural networks, we show in Table 3 that the OpenCon is competitive for transformer-based ViT model (Dosovitskiy et al., 2020). We adopt the ViT-B/16 architecture with DINO pre-trained weights (Caron et al., 2021), following the pipeline used in Vaze et al. (2022). In Table 3, we compare OpenCon's performance with ORCA (Cao et al., 2022), GCD (Vaze et al., 2022), *k*-Means (MacQueen, 1967), RankStats+ (Zhao & Han, 2021) and UNO+ (Fini et al., 2021) on ViT-B-16 architecture. On ImageNet-100, we improve upon the best baseline by 9.9 in terms of overall accuracy.

Table 3: Comparison of accuracy on ViT-B/16 architecture (Dosovitskiy et al., 2020). Results are reported on ImageNet-100.

| Methods | All | Novel | Seen |
|---|---|---|---|
| ORCA (Cao et al., 2022) | 73.5 | 64.6 | 89.3 |
| GCD (Vaze et al., 2022) | 74.1 | 66.3 | 89.8 |
| *k*-Means (MacQueen, 1967) | 72.7 | 71.3 | 75.5 |
| RankStats+ (Zhao & Han, 2021) | 37.1 | 24.8 | 61.6 |
| UNO+ (Fini et al., 2021) | 70.3 | 57.9 | **95.0** |
| OpenCon (Ours) | **84.0** | **81.2** | 93.8 |

**OpenCon learns more distinguishable representations** We visualize feature embeddings using UMAP (McInnes et al., 2018) in Figure 2. Different colors represent different ground-truth class labels. For clarity, we use the ImageNet-100 dataset and visualize a subset of 10 classes. We can observe that OpenCon produces a better embedding space than GCD and ORCA. In particular, ORCA does not produce distinguishable representations for novel classes, especially when the number of classes increases. The features of GCD are improved, yet with some class overlapping (*e.g.*, two orange classes). For reader's reference, we also include the version with a subset of 20 classes in Appendix H, where OpenCon displays more distinguishable representations.

## 6 A Comprehensive Analysis of OpenCon

**Prototype-based OOD detection is important** In Figure 3, we ablate the contribution of a key component in OpenCon: prototype-based OOD detection (*c.f.* Section 3.2). To systematically analyze the effect, we report the performance under varying percentile $p \in \{0, 10, 30, 50, 70, 90\}$. Each $p$ corresponds to a different threshold $\lambda$ for separating known vs. novel data in $\mathcal{D}_u$. In the extreme case with $p = 0$, $\mathcal{D}_n$ becomes equivalent to $\mathcal{D}_u$, and hence the contrastive loss $\mathcal{L}_n$ is applied to the entire unlabeled data. We highlight two findings: (1) Without OOD detection ($p = 0$), the unseen accuracy reduces by 2.4%, compared to the best setting ($p = 70\%$). This affirms the importance of OOD detection for better representation learning. (2) A higher percentile $p$, in general, leads to better performance. We also provide theoretical insights in Appendix D showing OOD detection helps contrastive learning of novel classes by having fewer candidate classes.

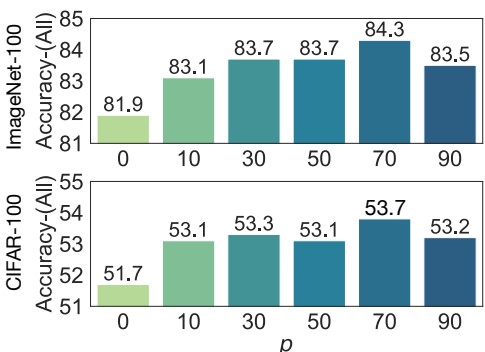

Figure 3: Effect of $p$ on ImageNet-100 and CIFAR-100, measured by the overall accuracy. $p = 0$ means no OOD detection.

Table 4: Ablation study on loss component.

| Loss Components | CIFAR-100 | | | ImageNet-100 | | |
|---|---|---|---|---|---|---|
| | All | Novel | Seen | All | Novel | Seen |
| w/o $\mathcal{L}_l$ | 43.3 | 47.1 | 38.8 | 68.6 | 73.4 | 59.1 |
| w/o $\mathcal{L}_u$ | 36.9 | 28.3 | 63.4 | 55.9 | 39.3 | 89.2 |
| w/o $\mathcal{L}_n$ | 46.6 | 42.2 | **70.3** | 74.5 | 70.7 | **91.0** |
| OpenCon (ours) | **52.7**$^{\pm 0.6}$ | **47.8**$^{\pm 0.6}$ | 69.1$^{\pm 0.3}$ | **83.8**$^{\pm 0.3}$ | **80.8**$^{\pm 0.3}$ | 90.6$^{\pm 0.1}$ |

**Ablation study on the loss components** Recall that our overall objective function in Equation 7 consists of three parts. We ablate the contributions of each component in Table 4. Specifically, we modify OpenCon by removing: (i) supervised objective (*i.e.*, w/o $\mathcal{D}_l$), (ii) unsupervised objective on the entire unlabeled data (*i.e.*, w/o $\mathcal{D}_u$), and (iii) prototype-based contrastive learning on novel data $\mathcal{D}_n$. We have the following key observations: (1) Both supervised objective $\mathcal{L}_l$ and unsupervised loss $\mathcal{L}_u$ are indispensable parts of open-world representation learning. This suits the complex nature of our training data, which requires learning on both labeled and unlabeled data, across both known and novel classes. (2) Purely combining SupCon (Khosla et al., 2020) and SimCLR (Chen et al., 2020a)—as used in GCD (Vaze et al., 2022)—does not give competitive results. For example, the overall accuracy is **9.3**% lower than our method on the ImageNet-100 dataset. In contrast, having $\mathcal{L}_n$ encourages closely aligned representations to *all* entries from the same *predicted* class, resulting in a more compact representation space for novel data. Overall, the ablation suggests that all losses in our framework work together synergistically to enhance the representation quality.

**Handling an unknown number of novel classes** In practice, we often do not know the number of classes $|\mathcal{Y}_{\text{all}}|$ in advance. This is the dilemma faced by Open-Con and other baselines as well. In such cases, one can apply OpenCon by first estimating the number of classes. For a fair comparison, we use the same estimation technique[2] as in Han et al. (2019); Cao et al. (2022). On CIFAR-100, the estimated total number of classes is 124. At the beginning of training, we initialize the same number of prototypes accordingly. Results in Table 5 show that OpenCon outperforms

Table 5: Accuracy on CIFAR-100 dataset with an unknown number of classes.

| Methods | All | Novel | Seen |
|---|---|---|---|
| ORCA (Cao et al., 2022) | 46.4 | 40.0 | 66.3 |
| GCD (Vaze et al., 2022) | 47.2 | 41.9 | **69.8** |
| OpenCon (Known $|\mathcal{Y}_{\text{all}}|$) | 53.7 | **48.7** | **69.0** |
| OpenCon (Unknown $|\mathcal{Y}_{\text{all}}|$) | **53.7** | 48.2 | 68.8 |

---

[2]A clustering algorithm is performed on the combination of labeled data and unlabeled data. The optimal number of classes is chosen by validating clustering accuracy on the labeled data.

the best baseline ORCA (Cao et al., 2022) by **7.3**%. Interestingly, despite the initial number of classes, the training process will converge to solutions that closely match the ground truth number of classes. For example, at convergence, we observe a total number of 109 actual clusters. The remaining ones have no samples assigned, hence can be discarded. Overall, with the estimated number of classes, OpenCon can achieve similar performance compared to the setting in which the number of classes is known.

**OpenCon is robust under a smaller number of labeled examples, and a larger number of novel classes** We show that OpenCon's strong performance holds under more challenging settings with: (1) reduced fractions of labeled examples, and (2) different ratios of known vs. novel classes. The results are summarized in Table 6. First, we reduce the labeling ratio from 50% (default) to 25% and 10%, while keeping the number of known classes to be the same (*i.e.*, 50). With fewer labeled samples in the known classes, the unlabeled sample set size will expand accordingly. It posits more challenges for novelty discovery and representation learning. On ImageNet-100, OpenCon substantially improves the novel class accuracy by **10**% compared to ORCA and GCD, when only 10% samples are labeled. Secondly, we further increase the number of novel classes, from 50 (default) to 75 and 90 respectively. On ImageNet-100 with 75 novel classes ($|\mathcal{Y}_l| = 25$), OpenCon improves the novel class accuracy by **16.7**% over ORCA (Cao et al., 2022). Overall our experiments confirm the robustness of OpenCon under various settings.

Table 6: Accuracy on CIFAR-100 under varying different labeling ratios (for labeled data) and different numbers of known classes ($|\mathcal{Y}_l|$). The number of novel classes is $100 - |\mathcal{Y}_l|$.

| Labeling Ratio | $|\mathcal{Y}_l|$ | Method | CIFAR-100 | | | ImageNet-100 | | |
|---|---|---|---|---|---|---|---|---|
| | | | All | Novel | Seen | All | Novel | Seen |
| 0.5 | 50 | ORCA | 47.0 | 41.3 | 66.2 | 76.6 | 69.0 | 88.9 |
| | | GCD | 47.2 | 43.6 | **69.4** | 75.1 | 73.2 | **90.9** |
| | | OpenCon | **53.7** | **48.7** | 69.0 | **84.3** | **81.1** | 90.7 |
| 0.25 | 50 | ORCA | 47.6 | 42.5 | 61.8 | 72.2 | 64.2 | 87.3 |
| | | GCD | 41.4 | 39.3 | **66.0** | 76.7 | 69.1 | 89.1 |
| | | OpenCon | **51.3** | **44.6** | 65.5 | **82.0** | **77.1** | **90.6** |
| 0.1 | 50 | ORCA | 41.2 | 37.7 | 54.6 | 68.7 | 56.8 | 83.4 |
| | | GCD | 37.0 | 38.6 | 62.2 | 69.4 | 56.6 | **85.8** |
| | | OpenCon | **48.2** | **44.4** | **62.5** | **75.4** | **66.8** | 85.2 |
| 0.5 | 25 | ORCA | 40.4 | 38.8 | 66.0 | 57.8 | 54.0 | 89.4 |
| | | GCD | 41.6 | 39.2 | 70.0 | 65.3 | 63.2 | 90.8 |
| | | OpenCon | **43.9** | **41.9** | **70.2** | **74.5** | **72.7** | **91.2** |
| 0.5 | 10 | ORCA | 34.3 | 33.8 | 67.4 | 45.1 | 43.6 | 93.0 |
| | | GCD | 38.4 | 36.8 | 61.3 | 53.3 | 52.9 | 94.2 |
| | | OpenCon | **40.9** | **40.5** | **69.9** | **59.0** | **58.2** | **94.3** |

## 7 Related Work

**Contrastive learning** A great number of works have explored the effectiveness of contrastive loss in unsupervised representation learning: InfoNCE (Van den Oord et al., 2018), SimCLR (Chen et al., 2020a), SWaV (Caron et al., 2020), MoCo (He et al., 2020), SEER (Goyal et al., 2021) and  (Li et al., 2020; 2021b; Zhang et al., 2021). It motivates follow-up works to on weakly supervised learning tasks (Zheng et al., 2021; Tsai et al., 2022), semi-supervised learning (Chen et al., 2020b; Li et al., 2021a; Zhang et al., 2022b; Yang et al., 2022a), supervised learning with noise  (Wu et al., 2021; Karim et al., 2022; Li et al., 2022a), continual learning (Cha et al., 2021), long-tailed recognition (Cui et al., 2021; Tian et al., 2021; Jiang et al., 2021; Li et al., 2022b), few-shot learning (Gao et al., 2021), partial label learning (Wang et al., 2022), novel class discovery (Zhong et al., 2021a; Zhao & Han, 2021; Fini et al., 2021), hierarchical multi-label learning (Zhang et al., 2022a). Under different circumstances, all works adopt different choices of the positive set, which is not limited to the self-augmented view in SimCLR (Chen et al., 2020a). Specifically, with label information available, SupCon (Khosla et al., 2020) improved representation quality by aligning features within the same

class. Without supervision for the unlabeled data, Dwibedi et al. (2021) used the nearest neighbor as positive pair to learn a compact embedding space. Different from prior works, we focus on the open-world representation learning problem, which is largely unexplored.

**Out-of-distribution detection** The problem of classification with rejection can date back to early works which considered simple model families such as SVMs (Fumera & Roli, 2002). In deep learning, the problem of out-of-distribution (OOD) detection has received significant research attention in recent years. Yang et al. (2022b) survey a line of works, including confidence-based methods (Bendale & Boult, 2016; Hendrycks & Gimpel, 2017; Liang et al., 2018; Huang & Li, 2021), energy-based score (Lin et al., 2021; Liu et al., 2020; Wang et al., 2021; Sun et al., 2021; Sun & Li, 2022; Morteza & Li, 2022), distance-based approaches (Lee et al., 2018; Tack et al., 2020; Sehwag et al., 2021; Sun et al., 2022; Ming et al., 2022a;b), gradient-based score (Huang et al., 2021b), and Bayesian approaches (Gal & Ghahramani, 2016; Lakshminarayanan et al., 2017; Maddox et al., 2019; Malinin & Gales, 2018; 2019). In our framework, OOD detection is based on the distance to the prototype of the closest known class. We show in Appendix J that several popular OOD detection methods give similar performance.

**Novel category discovery** At an earlier stage, the problem of novel category discovery (NCD) is targeted as a transfer learning problem in DTC (Han et al., 2019), KCL (Hsu et al., 2018), MCL (Hsu et al., 2019). The learning is generally in a two-stage manner: the model is firstly trained with the labeled data and then transfers knowledge to learn the unlabeled data. OpenMix (Zhong et al., 2021b) further proposes an end-to-end framework by mixing the seen and novel classes in a joint space. In recent studies, many researchers incorporate representation learning for NCD like RankStats (Zhao & Han, 2021), NCL (Zhong et al., 2021a) and UNO (Fini et al., 2021). In our setting, the unlabeled test set consists of novel classes but also classes previously seen in the labeled data that need to be separated.

**Semi-supervised learning** A great number of early works (Chapelle et al., 2006; Lee et al., 2013; Sajjadi et al., 2016; Laine & Aila, 2017; Zhai et al., 2019; Rebuffi et al., 2020; Kurakin et al., 2020) have been proposed to tackle the problem of semi-supervised learning (SSL). Typically, a standard cross-entropy loss is applied to the labeled data, and a consistency loss (Laine & Aila, 2017; Kurakin et al., 2020) or self-supervised loss (Sajjadi et al., 2016; Zhai et al., 2019; Rebuffi et al., 2020) is applied to the unlabeled data. Under the closed-world assumption, SSL methods achieve competitive performance which is close to the supervised methods. Later works (Oliver et al., 2018; Chen et al., 2020c) point out that including novel classes in the unlabeled set can downgrade the performance. In Guo et al. (2020); Chen et al. (2020c); Yu et al. (2020); Park et al. (2021); Saito et al. (2021); Huang et al. (2021a); Yang et al. (2022a), OOD detection techniques are wielded to separate the OOD samples in the unlabeled data. Recent works (Vaze et al., 2022; Cao et al., 2022; Rizve et al., 2022) further require the model to group samples from novel classes into semantically meaningful clusters. In our framework, we unify the novelty class detection and the representation learning and achieve competitive performance.

## 8    Conclusion

This paper provides a new learning framework, *open-world contrastive learning* (OpenCon) that learns highly distinguishable representations for both known and novel classes in an open-world setting. Our open-world setting can generalize traditional representation learning and offers stronger flexibility. We provide important insights that the separation between known vs. novel data in the unlabeled data and the pseudo supervision for data in novel classes is critical. Extensive experiments show that OpenCon can notably improve the accuracy on both known and novel classes compared to the current best method ORCA. As a shared challenge by all methods, one limitation is that the prototype number in our end-to-end training framework needs to be pre-specified. An interesting future work may include the mechanism to dynamically estimate and adjust the class number during the training stage. The broader impact statement is included Appendix A.

## Acknowledgement

The authors would also like to thank Haobo Wang and TMLR reviewers for the helpful suggestions and feedback.

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

**Appendix**
**OpenCon: Open-world Contrastive Learning**



## A   Broader Impacts

Our project aims to improve representation learning in the open-world setting. Our study leads to direct benefits and societal impacts for real-world applications such as e-commerce which may encounter brand-new products together with known products. Our study does not involve any human subjects or violation of legal compliance. In some rare cases, the algorithm may identify unlabeled minority sub-groups if applied to images of humans or reflect biases in the data collection. Through our study and releasing our code, we hope that our work will inspire more future works to tackle this important problem.

## B   Preliminaries of Supervised Contrastive Loss and Self-supervised Contrastive Loss

Recall in the main paper, we provide a general form of the per-sample contrastive loss:

$$
\mathcal{L}_\phi\big(\mathbf{x}; \tau, \mathcal{P}(\mathbf{x}), \mathcal{N}(\mathbf{x})\big) = -\frac{1}{|\mathcal{P}(\mathbf{x})|} \sum_{\mathbf{z}^+ \in \mathcal{P}(\mathbf{x})} \log \frac{\exp(\mathbf{z}^\top \cdot \mathbf{z}^+/\tau)}{\sum_{\mathbf{z}^- \in \mathcal{N}(\mathbf{x})} \exp(\mathbf{z}^\top \cdot \mathbf{z}^-/\tau)},
$$

where $\tau$ is the temperature parameter, $\mathbf{z}$ is the $L_2$ normalized embedding of $\mathbf{x}$, $\mathcal{P}(\mathbf{x})$ is the positive set of embeddings $w.r.t.$ $\mathbf{z}$, and $\mathcal{N}(\mathbf{x})$ is the negative set of embeddings.

In this section, we provide a detailed definition of Supervised Contrastive Loss (SupCon) (Khosla et al., 2020) and Self-supervised Contrastive Loss (SimCLR) (Chen et al., 2020a).

**Supervised Contrastive Loss**  For a mini-batch $\mathcal{B}_l$ with samples drawn from $\mathcal{D}_l$, we apply two random augmentations for each sample and generate a multi-viewed batch $\tilde{\mathcal{B}}_l$. We denote the embeddings of the multi-viewed batch as $\mathcal{A}_l$, where the cardinality $|\mathcal{A}_l| = 2|\mathcal{B}_l|$. For any sample $\mathbf{x}$ in the mini-batch $\tilde{\mathcal{B}}_l$, the positive and negative set of embeddings are as follows:

$$
\mathcal{P}_l(\mathbf{x}) = \{\mathbf{z}' \mid \mathbf{z}' \in \{\mathcal{A}_l\backslash\mathbf{z}\}, y' = y\}
$$
$$
\mathcal{N}_l(\mathbf{x}) = \mathcal{A}_l\backslash\mathbf{z},
$$

where $y$ is the ground-truth label of $\mathbf{x}$, and $y'$ is the predicted label for the corresponding sample of $\mathbf{z}'$. Formally, the supervised contrastive loss is defined as:

$$
\mathcal{L}_l = \mathcal{L}_\phi\big(\mathbf{x}; \tau_l, \mathcal{P}_l(\mathbf{x}), \mathcal{N}_l(\mathbf{x})\big),
$$

where $\tau_l$ is the temperature.

**Self-Supervised Contrastive Loss**  For a mini-batch $\mathcal{B}_u$ with samples drawn from unlabeled dataset $\mathcal{D}_u$, we apply two random augmentations for each sample and generate a multi-viewed batch $\tilde{\mathcal{B}}_u$. We denote the embeddings of the multi-viewed batch as $\mathcal{A}_u$, where the cardinality $|\mathcal{A}_u| = 2|\mathcal{B}_u|$. For any sample $\mathbf{x}$ in the mini-batch $\tilde{\mathcal{B}}_u$, the positive and negative set of embeddings is as follows:

$$
\mathcal{P}_u(\mathbf{x}) = \{\mathbf{z}' \mid \mathbf{z}' = \phi(\mathbf{x}'), \mathbf{x}' \text{ is augmented from the same sample as } \mathbf{x}\}
$$
$$
\mathcal{N}_u(\mathbf{x}) = \mathcal{A}_u\backslash\mathbf{z}
$$

The self-supervised contrastive loss is then defined as:

$$\mathcal{L}_u = \mathcal{L}_\phi\big(\mathbf{x}; \tau_u, \mathcal{P}_u(\mathbf{x}), \mathcal{N}_u(\mathbf{x})\big),$$

where $\tau_u$ is the temperature.

## C   Algorithm

Below we summarize the full algorithm of open-world contrastive learning. The notation of $\mathcal{B}_u$, $\mathcal{B}_l$, $\mathcal{A}_u$, $\mathcal{A}_l$ is defined in Appendix B.

---

**Algorithm 1** Open-world Contrastive Learning

---

**Input:** Labeled set $\mathcal{D}_l = \{\mathbf{x}_i, y_i\}_{i=1}^n$ and unlabeled set $\mathcal{D}_u = \{\mathbf{x}_i\}_{i=1}^m$, neural network encoder $\phi$, randomly initialized prototypes $\mathbf{M}$.
**Training Stage**:
**repeat**
  **Data Preparation**:
  Sample a mini-batch of labeled data $\mathcal{B}_l = \{\mathbf{x}_i, y_i\}_{i=1}^{b_l}$ and unlabeled data $\mathcal{B}_u = \{\mathbf{x}_i\}_{i=1}^{b_u}$
  Generate augmented batch and extract normalized embedding set $\mathcal{A}_l, \mathcal{A}_u$
  **OOD detection**:
  Calculate OOD detection threshold $\lambda$ by $\mathcal{A}_l$
  Separate $\mathcal{A}_n$ from $\mathcal{A}_u$
  **Positive/Negative Set Selection**:
  Assign pseudo-labels $\hat{y}_i$ by prototypes for each sample in $\mathcal{A}_n$
  Obtain $\mathcal{P}_n, \mathcal{N}_n$ from $\mathcal{A}_n$
  **Back-propagation**:
  Calculate loss $\mathcal{L}_{\text{OpenCon}}$
  Update network $\phi$ using the gradients.
  **Prototype Update**:
  Update prototype vectors with Equation 6
**until** Convergence

---

## D   Theoretical Justification of OOD Detection for OpenCon

In this section, we theoretically show that OOD detection helps open-world representation learning by reducing the lower bound of loss $\mathbb{E}_{\mathbf{x} \in \mathcal{D}_n} \mathcal{L}_n(\mathbf{x})$. We start with the definition of the supervised loss of the Mean Classifier, which provides the lower bound in Lemma D.2.

**Definition D.1.** (Mean Classifier) the mean classifier is a linear layer with weight matrix $\mathbf{M}^*$ whose $c$-th row is the mean $\tilde{\boldsymbol{\mu}}_c$ of representations of inputs with class $c$: $\tilde{\boldsymbol{\mu}}_c = \mathbb{E}_{\mathbf{x} \in \mathcal{S}(c)}[\phi(\mathbf{x})]$, where $\mathcal{S}(c)$ defined in Appendix 4 is the set of samples with predicted label $c$. The average supervised loss of its mean classifier is:

$$\mathcal{L}_{sup}^* := - \mathbb{E}_{c^+, c^- \in \mathcal{Y}_{\text{all}}^2} \left[ \mathbb{E}_{\mathbf{x} \in \mathcal{S}(c^+)} \phi(\mathbf{x})(\tilde{\boldsymbol{\mu}}_{c^+} - \tilde{\boldsymbol{\mu}}_{c^-}) \mid c^+ \neq c^- \right] \tag{9}$$

**Lemma D.2.** *Let $\gamma = p(c^+ = c^-), c^+, c^- \in \mathcal{Y}_{all}^2$, it holds that*

$$\mathbb{E}_{\mathbf{x} \in \mathcal{D}_n} \mathcal{L}_n(\mathbf{x}) \geq \frac{1-\gamma}{\tau} \mathcal{L}_{sup}^*$$

*Proof.*

$$
\mathop{\mathbb{E}}_{\mathbf{x}\in\mathcal{D}_n}\mathcal{L}_n(\mathbf{x}) = \mathop{\mathbb{E}}_{\mathbf{x}\in\mathcal{D}_n} - \frac{1}{|\mathcal{P}(\mathbf{x})|}\sum_{\mathbf{z}^+\in\mathcal{P}(\mathbf{x})}\log\frac{\exp(\mathbf{z}^\top\cdot\mathbf{z}^+/\tau)}{\sum_{\mathbf{z}^-\in\mathcal{N}(\mathbf{x})}\exp(\mathbf{z}\cdot\mathbf{z}^-/\tau)}
$$

$$
= \mathop{\mathbb{E}}_{\mathbf{x}\in\mathcal{D}_n}\left[-\frac{1}{|\mathcal{P}(\mathbf{x})|}\sum_{\mathbf{z}^+\in\mathcal{P}(\mathbf{x})}(\mathbf{z}^\top\cdot\mathbf{z}^+/\tau) + \frac{1}{|\mathcal{P}(\mathbf{x})|}\sum_{\mathbf{z}^+\in\mathcal{P}(\mathbf{x})}\log\sum_{\mathbf{z}^-\in\mathcal{N}(\mathbf{x})}\exp(\mathbf{z}^\top\cdot\mathbf{z}^-/\tau)\right]
$$

$$
\stackrel{(a)}{\approx} - \mathop{\mathbb{E}}_{c^+\in\mathcal{Y}_{\text{all}}}\mathop{\mathbb{E}}_{\mathbf{x},\mathbf{x}^+\in\mathcal{S}^2(c^+)}\left[\phi(\mathbf{x})^\top\cdot\phi(\mathbf{x}^+)/\tau - \log\mathop{\mathbb{E}}_{c^-\in\mathcal{Y}_{\text{all}},\mathbf{x}\in\mathcal{S}(c^-)}\exp(\phi(\mathbf{x})^\top\cdot\phi(\mathbf{x}^-)/\tau)\right]
$$

$$
\stackrel{(b)}{\geq} - \mathop{\mathbb{E}}_{c^+\in\mathcal{Y}_{\text{all}}}\mathop{\mathbb{E}}_{\mathbf{x},\mathbf{x}^+\in\mathcal{S}^2(c^+)}\left[\phi(\mathbf{x})^\top\cdot\phi(\mathbf{x}^+)/\tau - \mathop{\mathbb{E}}_{c^-\in\mathcal{Y}_{\text{all}},\mathbf{x}\in\mathcal{S}(c^-)}\phi(\mathbf{x})^\top\cdot\phi(\mathbf{x}^-)/\tau\right]
$$

$$
= - \mathop{\mathbb{E}}_{c^+,c^-\in\mathcal{Y}_{\text{all}}}\mathop{\mathbb{E}}_{\mathbf{x}\in\mathcal{S}(c^+)}\phi(\mathbf{x})(\tilde{\boldsymbol{\mu}}_{c^+}-\tilde{\boldsymbol{\mu}}_{c^-})/\tau
$$

$$
= p(c^+\neq c^-)\cdot\mathop{\mathbb{E}}_{c^+,c^-\in\mathcal{Y}_{\text{all}}^2}\left[-\mathop{\mathbb{E}}_{\mathbf{x}\in\mathcal{S}(c^+)}\phi(\mathbf{x})(\tilde{\boldsymbol{\mu}}_{c^+}-\tilde{\boldsymbol{\mu}}_{c^-})/\tau \mid c^+\neq c^-\right] + p(c^+\neq c^-)\cdot 0
$$

$$
= \frac{1-\gamma}{\tau}\mathcal{L}_{sup}^*,
$$

where in (a) we approximate the summation over the positive/negative set by taking the expectation over the positive/negative sample in set $\mathcal{S}(c)$ (defined in Appendix 4) and in (b) we apply the Jensen Inequality since the log is a concave function. □

In the first step, we show in Lemma D.2 that $\mathop{\mathbb{E}}_{\mathbf{x}\in\mathcal{D}_n}\mathcal{L}_n(\mathbf{x})$ is lower-bounded by a constant $\frac{1-\gamma}{\tau}$ times supervised loss $\mathcal{L}_{sup}^*$ defined in Definition D.1. Note that $\mathcal{L}_{sup}^*$ is non-positive and close to $-1$ in practice. Then the lower bound of $\mathop{\mathbb{E}}_{\mathbf{x}\in\mathcal{D}_n}\mathcal{L}_n(\mathbf{x})$ has a positive correlation with $\gamma = p(c^+ = c^-)$. Note that $\gamma$ can be reduced by OOD detection. To explain this:

When we separate novelty samples and form $\mathcal{D}_n$, it has fewer hidden classes than $\mathcal{D}_u$. With fewer hidden classes, the probability of $c^+$ being equal to $c^-$ in random sampling is decreased, and thus reduces the lower bound of the $\mathop{\mathbb{E}}_{\mathbf{x}\in\mathcal{D}_n}\mathcal{L}_n(\mathbf{x})$.

In summary, OOD detection facilitates open-world contrastive learning by having fewer candidate classes.

# E    Discussion on Using Samples in $\mathcal{D}_u\backslash\mathcal{D}_n$

We discussed in Section 3.2 that samples from $\mathcal{D}_u\backslash\mathcal{D}_n$ contain indistinguishable data from known and novel classes. In this section, we show that using these samples for prototype-based learning may be undesirable.

We first show that the overlapping between the novel and known classes in $\mathcal{D}_u\backslash\mathcal{D}_n$ can be non-trivial. In Figure 4, we show the distribution plot of the scores $\max_{j\in\mathcal{Y}_l}\boldsymbol{\mu}_j^\top\cdot\phi(\mathbf{x}_i)$. It is notable that there exists a large overlapping area when $\max_{j\in\mathcal{Y}_l}\boldsymbol{\mu}_j^\top\cdot\phi(\mathbf{x}_i) > \lambda$. For visualization clarity, we color the known classes in blue and the novel classes in gray.

We next show that using this part of data will be harmful to representation learning. Specifically, we replace $\mathcal{L}_l$ to be the following loss:

$$
\mathcal{L}_{known} = \sum_{\mathbf{x}\in\tilde{\mathcal{B}}_k}\mathcal{L}_\phi\left(\mathbf{x};\tau_k,\mathcal{P}_k(\mathbf{x}),\mathcal{N}_k(\mathbf{x})\right),
$$

where we define $\mathcal{B}_k$ to be a minibatch with samples drawn from $\mathcal{D}_l\cup(\mathcal{D}_u\backslash\mathcal{D}_n)$—labeled data with known classes, along with the unlabeled data *predicted as known classes*. And we apply two random augmentations for each sample and generate a multi-viewed batch $\tilde{\mathcal{B}}_k$. We denote the embeddings of the multi-viewed batch

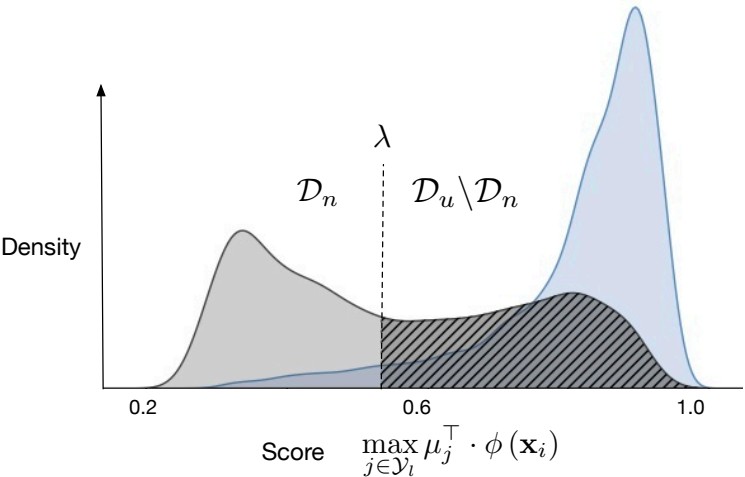

Figure 4: Distribution of OOD score $\max_{j\in\mathcal{Y}_l}\boldsymbol{\mu}_j^\top\cdot\phi(\mathbf{x}_i)$ for unlabeled data $\mathcal{D}_u$ on CIFAR-100. For visual clarity, gray indicates samples from novel classes and blue indicates samples from known classes. The shaded area highlights samples from novel classes but misidentified as known classes.

as $\mathcal{A}_k$. The positive set of embeddings is as follows:

$$\mathcal{P}_k(\mathbf{x}) = \{\mathbf{z}' \mid \mathbf{z}' \in \{\mathcal{A}_k\backslash\mathbf{z}\}, \tilde{y}' = \tilde{y}\}$$
$$\mathcal{N}_k(\mathbf{x}) = \mathcal{A}_k\backslash\mathbf{z},$$

where

$$\tilde{y}_i = \begin{cases} \hat{y}_i \text{ (pseudo-label)} & \text{if } \mathbf{x}_i \in \mathcal{D}_u \\ y_i \text{ (ground-truth label)}, & \text{if } \mathbf{x}_i \in \mathcal{D}_l \end{cases}$$

Intuitively, $\mathcal{L}_{known}$ is an extension of $\mathcal{L}_l$, where we utilize both labeled and unlabeled data from known classes for representation learning. The final loss now becomes:

$$\mathcal{L}_{\text{Modified}} = \lambda_n\mathcal{L}_n + \lambda_k\mathcal{L}_{known} + \lambda_u\mathcal{L}_u, \tag{10}$$

We show results in Table 7. Compared to the original loss, the seen accuracy drops by 6.6%. This finding suggests that using $\mathcal{D}_u\backslash\mathcal{D}_n$ for prototype-based learning is suboptimal.

Table 7: Comparison with loss $\mathcal{L}_{\text{Modified}}$ on CIFAR-100.

| Method | CIFAR-100 | | |
| --- | --- | --- | --- |
| | **All** | **Novel** | **Seen** |
| $\mathcal{L}_{\text{Modified}}$ | 47.7 | 46.4 | 62.4 |
| $\mathcal{L}_{\text{OpenCon}}$ | **53.7** | **48.7** | **69.0** |

# F   Proof Details

**Proof of Lemma 4.2**

*Proof.*

$$\operatorname*{argmax}_{\phi,\mathbf{M}} \sum_{i=1}^{|\mathcal{D}_n|} \sum_{c\in\mathcal{Y}_{\text{all}}} q_i(c) \log \frac{p\left(\mathbf{x}_i, c|\phi, \mathbf{M}\right)}{q_i(c)} \overset{(a)}{=} \operatorname*{argmax}_{\phi,\mathbf{M}} \sum_{i=1}^{|\mathcal{D}_n|} \sum_{c\in\mathcal{Y}_{\text{all}}} q_i(c) \log p\left(\mathbf{x}_i|c, \phi, \mathbf{M}\right)$$

$$\overset{(b)}{=} \operatorname*{argmax}_{\phi,\mathbf{M}} \sum_{i=1}^{|\mathcal{D}_n|} \sum_{c\in\mathcal{Y}_{\text{all}}} \mathbf{1}\{\hat{y}_i = c\} \log p\left(\mathbf{x}_i|c, \phi, \mathbf{M}\right)$$

$$\overset{(d)}{=} \operatorname*{argmax}_{\phi,\mathbf{M}} \sum_{c\in\mathcal{Y}_{\text{all}}} \sum_{\mathbf{x}\in\mathcal{S}(c)} \log p(\mathbf{x}|c, \phi, \mathbf{M})$$

$$\overset{(e)}{=} \operatorname*{argmax}_{\phi,\mathbf{M}} \sum_{c\in\mathcal{Y}_{\text{all}}} \sum_{\mathbf{x}\in\mathcal{S}(c)} \phi(\mathbf{x})^\top \cdot \boldsymbol{\mu}_c,$$

where equation $(a)$ is given by removing the constant term $q_i(c) \log \frac{p(c)}{q_i(c)}$ in argmax, (b) is by plugging $q_i(c)$, (d) is by reorganizing the index, and (e) is by plugging the vMF density function and removing the constant. $\qquad \square$

**Proof of Lemma 4.3**

*Proof.*

$$\operatorname*{argmin}_{\phi} \sum_{\mathbf{x}\in\mathcal{D}_n} \mathcal{L}_a(\mathbf{x}) = \operatorname*{argmin}_{\phi} -\sum_{\mathbf{x}\in\mathcal{D}_n} \frac{1}{|\mathcal{P}(\mathbf{x})|} \sum_{\mathbf{z}^+\in\mathcal{P}(\mathbf{x})} \phi(\mathbf{x})^\top \cdot \mathbf{z}^+$$

$$= \operatorname*{argmin}_{\phi} -\sum_{c\in\mathcal{Y}_{\text{all}}} \sum_{\mathbf{x}\in\mathcal{S}(c)} \frac{1}{|\mathcal{S}_c|-1} \sum_{\boldsymbol{x}^+\in\mathcal{S}(c)\backslash\mathbf{x}} \phi(\mathbf{x})^\top \cdot \phi(\mathbf{x}^+)$$

$$= \operatorname*{argmin}_{\phi} -\sum_{c\in\mathcal{Y}_{\text{all}}} \sum_{\mathbf{x}\in\mathcal{S}(c)} \frac{1}{|\mathcal{S}_c|-1} \left( \left( \sum_{\boldsymbol{x}^+\in\mathcal{S}(c)} \phi(\mathbf{x})^\top \cdot \phi(\mathbf{x}^+) \right) - 1 \right)$$

$$\overset{(a)}{=} \operatorname*{argmin}_{\phi} -\sum_{c\in\mathcal{Y}_{\text{all}}} \sum_{\mathbf{x}\in\mathcal{S}(c)} \eta_{\mathcal{S}_c} \phi(\mathbf{x})^\top \cdot \boldsymbol{\mu}_c^*$$

$$\overset{(b)}{\approx} \operatorname*{argmax}_{\phi} \sum_{c\in\mathcal{Y}_{\text{all}}} \sum_{\mathbf{x}\in\mathcal{S}(c)} \phi(\mathbf{x})^\top \cdot \boldsymbol{\mu}_c^*,$$

where in (a) $\eta_{\mathcal{S}_c} = \frac{|\mathcal{S}_c|}{|\mathcal{S}_c|-1} \|\mathbb{E}_{\mathbf{x}\in\mathcal{S}(c)}[\phi(\mathbf{x})]\|_2$ is a constant value close to 1, and in (b) we show the approximation to the optimization target in Equation 8 with the fixed prototypes. The proof is done by using the equation in Lemma 4.2. $\qquad \square$

## G   Results on CIFAR-10

We show results for CIFAR-10 in Table 8, where OpenCon consistently outperforms strong baselines, particularly ORCA and GCD. Classes are divided into 50% known and 50% novel classes. We then select 50% of known classes as the labeled dataset and the rest as the unlabeled set. The division is consistent with (Cao et al., 2022), which allows us to compare the performance in a fair setting.

## H   More Qualitative Comparisons of Embeddings

In Figure 5, we visualize the feature embeddings for a subset of 20 classes using UMAP (McInnes et al., 2018). This covers more classes than what has been shown in the main paper (Figure 2). The model is trained on ImageNet-100. OpenCon produces a more compact and distinguishable embedding space than GCD and ORCA.

Table 8: Results on CIFAR-10. Asterisk ($\star$) denotes that the original method can not recognize seen classes. Dagger ($\dagger$) denotes the original method can not detect novel classes (and we had to extend it). Results on GCD, ORCA, and OpenCon (mean and standard deviation) are averaged over five different runs. The ORCA results are reported by running the official repo (Cao et al.).

| Method | CIFAR-10 | | |
|---|---|---|---|
| | **All** | **Novel** | **Seen** |
| $\dagger$**FixMatch** (Kurakin et al., 2020) | 49.5 | 50.4 | 71.5 |
| $\dagger$**DS$^3$L** (Guo et al., 2020) | 40.2 | 45.3 | 77.6 |
| $\dagger$**CGDL** (Sun et al., 2020) | 39.7 | 44.6 | 72.3 |
| $\star$**DTC** (Han et al., 2019) | 38.3 | 39.5 | 53.9 |
| $\star$**RankStats** (Zhao & Han, 2021) | 82.9 | 81.0 | 86.6 |
| $\star$**SimCLR** (Chen et al., 2020a) | 51.7 | 63.4 | 58.3 |
| **ORCA** (Cao et al., 2022) | $88.3^{\pm0.3}$ | $87.5^{\pm0.2}$ | $89.9^{\pm0.4}$ |
| **GCD** (Vaze et al., 2022) | $87.5^{\pm0.5}$ | $86.7^{\pm0.4}$ | $\mathbf{90.1}^{\pm0.3}$ |
| **OpenCon (Ours)** | $\mathbf{90.4}^{\pm0.6}$ | $\mathbf{91.1}^{\pm0.1}$ | $89.3^{\pm0.2}$ |

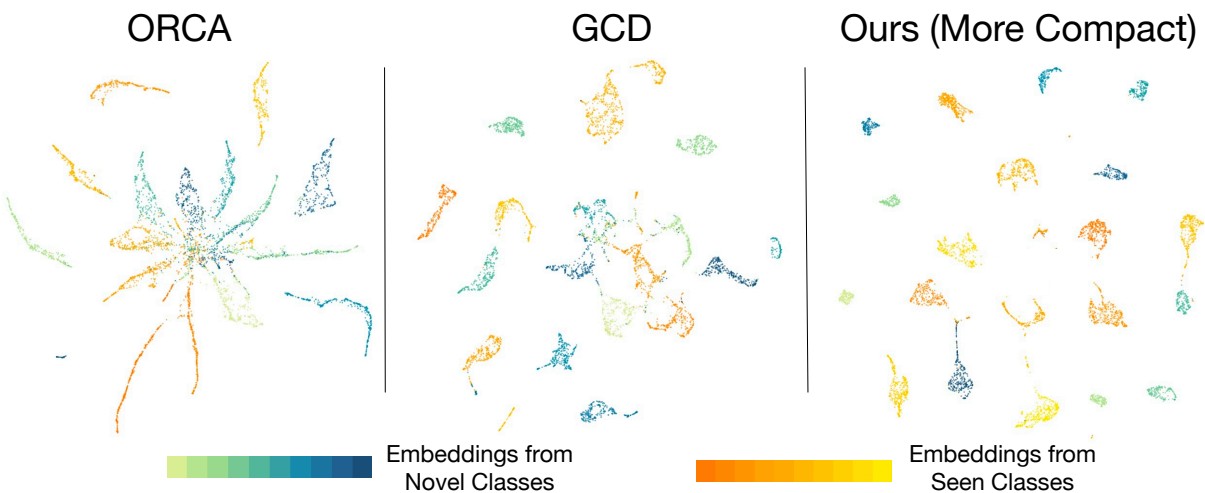

Figure 5: UMAP (McInnes et al., 2018) visualization of the feature embedding from 20 classes (10 for seen, 10 for novel). The model is trained on ImageNet-100 with ORCA (Cao et al., 2022), GCD (Vaze et al., 2022), and OpenCon (ours).

# I  Hyperparameters and Sensitivity Analysis

In this section, we introduce the hyper-parameter settings for OpenCon. We also show a validation strategy to determine important hyper-parameters (weight and temperature) in loss $\mathcal{L}_{OpenCon}$ and conduct a sensitivity analysis to show that the validation strategy can select near-optimal hyper-parameters. We start by introducing the basic training setting.

For CIFAR-100/ImageNet-100, the model is trained for 200/120 epochs with batch-size 512 using stochastic gradient descent with momentum 0.9, and weight decay $10^{-4}$. The learning rate starts at 0.02 and decays by a factor of 10 at the 50% and the 75% training stage. The momentum for prototype updating $\gamma$ is fixed at 0.9. The percentile $p$ for OOD detection is 70%. We fix the weight for the KL-divergence regularizer to be 0.05.

Since the label for $\mathcal{D}_u$ is not available, we propose a validation strategy by using labeled data $\mathcal{D}_l$. Specifically, we split the classes in $\mathcal{Y}_l$ equally into two parts: known classes and "novel" classes (for which we know the labels). Moreover, 50% samples of the selected known classes are labeled. We further use the new validation dataset to select the best hyper-parameters by grid searching. The selected hyper-parameter groups are

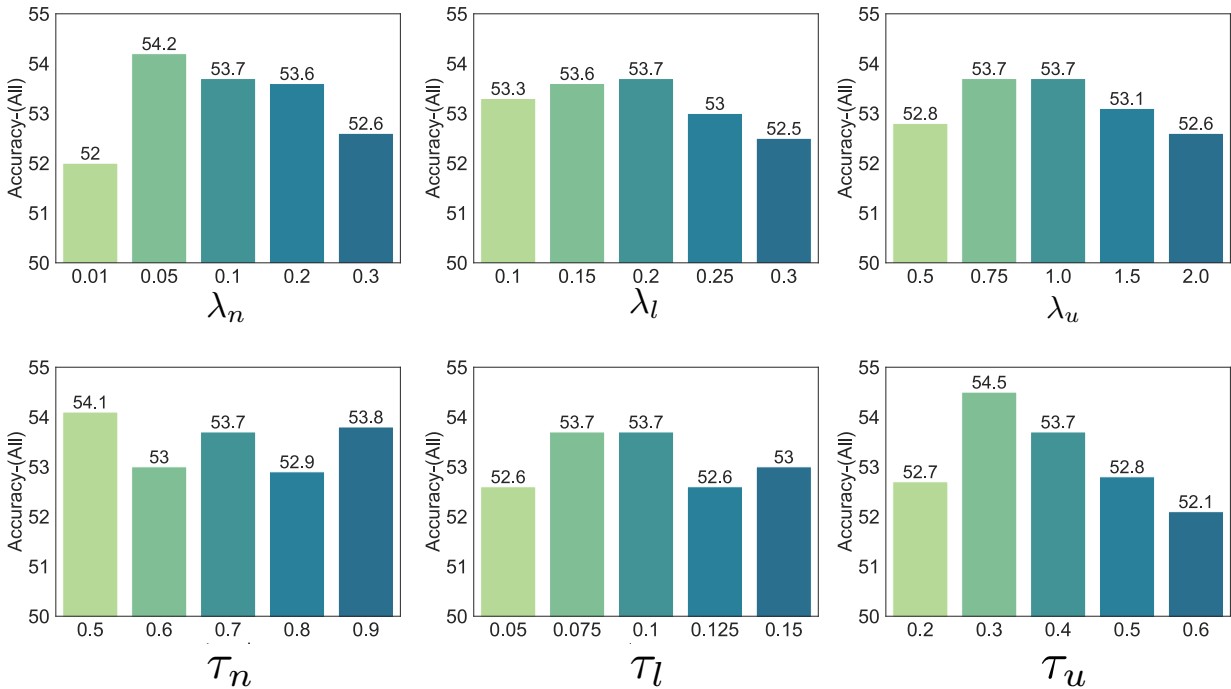

Figure 6: Sensitivity analysis of hyper-parameters on CIFAR-100. The overall accuracy is reported on three (weight, temperature) pairs: $(\lambda_n, \tau_n)$ for loss $\mathcal{L}_n$, $(\lambda_l, \tau_l)$ for loss $\mathcal{L}_l$, and the $(\lambda_u, \tau_u)$ for loss $\mathcal{L}_u$. The middle bar in each plot corresponds to the hyperparameter value used in our main experiments.

summarized in Table 9. Note that the only difference between CIFAR-100 and ImageNet-100 settings is the temperature of the self-supervised loss $\mathcal{L}_u$.

We show the sensitivity of hyper-parameters in Figure 6. The performance comparison in the bar plot for each hyper-parameter is reported by fixing other hyper-parameters. We see that our validation strategy successfully selects $\lambda_l$, $\lambda_u$, and $\tau_l$ with the optimal one, and the other three $\lambda_n$, $\tau_u$ and $\tau_n$ are close to the optimal (with <1% gap in overall accuracy).

Table 9: Hyperparameters in OpenCon.

|  | $\lambda_n$ | $\tau_n$ | $\lambda_l$ | $\tau_l$ | $\lambda_u$ | $\tau_u$ |
|---|---|---|---|---|---|---|
| ImageNet-100 | 0.1 | 0.7 | 0.2 | 0.1 | 1 | 0.6 |
| CIFAR-100 | 0.1 | 0.7 | 0.2 | 0.1 | 1 | 0.4 |

## J    OOD Detection Comparison

We compare different OOD detection methods in Table 10. Results show that several popular OOD detection methods produce similar OOD detection performance. Note that Mahalanobis (Lee et al., 2018) require heavier computation which causes an unbearable burden in the training stage. Our method incurs minimal computational overhead.

Table 10: Comparison of OOD detection performance with popular methods. Results are reported on CIFAR-100. Samples from 50 known classes are treated as in-distribution (ID) data and samples from the remaining 50 classes are used as out-of-distribution (OOD) data. The OOD detection threshold is estimated on the labeled known classes $\mathcal{D}_l$.

| Method | FPR95 | AUROC |
|---|---|---|
| MSP (Hendrycks & Gimpel, 2017) | 58.9 | 86.0 |
| Energy (Liu et al., 2020) | 57.1 | 87.4 |
| Mahalanobis (Lee et al., 2018) | 54.6 | 88.7 |
| Ours | 57.1 | 87.4 |

## K   Sensitivity Analysis on Estimated Class Number

In this section, we provide the sensitivity analysis of the estimated class number on CIFAR-100. We noticed that when the estimated class number is much larger than the actual class number (100), some prototypes will never be predicted by any of the training samples. So the training process will converge to solutions that closely match the ground truth number of classes. We show in Table 11 comparing the initial class number and the class number after converging. It shows that an overestimation of the class number has a negligible effect on the final converged class number.

Table 11: Converged class numbers under different initial class numbers.

| Initial Class Number | 100 | 110 | 120 | 130 | 150 | 200 |
|---|---|---|---|---|---|---|
| Converged Class Number | 100 | 106 | 108 | 109 | 109 | 109 |

## L   Performance on ImageNet-1k

To verify the effectiveness of OpenCon in a large-scale setting, we provide the results on ImageNet-1k in Table 12. The experiments are based on 500 known classes and 500 novel classes, with a labeling ratio = 0.5 for known classes. We notice that ORCA's performance degrades significantly in a large-scale setting. OpenCon significantly outperforms the best baseline GCD by 8.8%, in terms of overall accuracy.

Table 12: Comparison of classification accuracy on ImageNet-1k.

| Method | ImageNet-1k | | |
|---|---|---|---|
| | All | Novel | Seen |
| ORCA | 17.6 | 20.5 | 35.3 |
| GCD | 35.8 | 36.9 | 69.0 |
| OpenCon (ours) | **44.6** | **42.5** | **73.9** |

## M   Differences w.r.t. Prototypical Contrastive Learning

Our work is inspired by yet differs substantially from unsupervised prototypical contrastive learning (PCL) (Li et al., 2021b), in terms of problem setup (Section 2), learning objective (Section 3), and theoretical insights (Section 4). Importantly, PCL relies entirely on the unlabeled data $\mathcal{D}_u$ and does not leverage the availability of the labeled dataset. Instead, we consider the open-world semi-supervised learning setting (Cao et al., 2022), and leverage both labeled data (from known classes) and unlabeled data (from both known and novel classes). This setting leads to unique challenges that were not considered in PCL, such as how to learn strong representations for both the known and novel classes in a unified framework; and how to perform OOD

detection to separate known and novel samples in the unlabeled data. OpenCon makes new contributions by addressing these challenges in an end-to-end trainable framework, supported by theoretical analysis.

In Table 13, we show evidence that the PCL does not trivially apply in this open-world setting, and under-performs OpenCon.

Table 13: Comparison with PCL.

| Method | CIFAR-100 | | |
|---|---|---|---|
| | All | Novel | Seen |
| PCL (Li et al., 2021b) | 42.1 | 44.5 | 39.0 |
| OpenCon (Ours) | **53.7** | **48.7** | **69.0** |

## N  Experiments on Fine-grained Datasets

The fine-grained task is more challenging since the known and novel classes can be similar. We provide the results of the fine-grained task on Stanford Cars (Krause et al., 2013), CUB-200 (Vaze et al., 2021) and Herbarium19 (Tan et al., 2019) in Table 14. We adopt the ViT-B/16 architecture with DINO pre-trained weights (Caron et al., 2021). Results show that OpenCon outperforms the best baseline GCD by 10.1% on the Stanford Cars dataset, in terms of overall accuracy.

Table 14: Comparison on fine-grained datasets.

| Method | CUB-200 | | | Stanford Cars | | | Herbarium19 | | |
|---|---|---|---|---|---|---|---|---|---|
| | All | Novel | Seen | All | Novel | Seen | All | Novel | Seen |
| $k$-Means (MacQueen, 1967) | 34.3 | 32.1 | 38.9 | 13.8 | 10.6 | 12.8 | 12.9 | 12.8 | 12.9 |
| RankStats+ (Zhao & Han, 2021) | 33.3 | 24.2 | 51.6 | 28.3 | 12.1 | 61.8 | 27.9 | 12.8 | 55.8 |
| UNO+ (Fini et al., 2021) | 35.1 | 28.1 | 49.0 | 35.5 | 18.6 | 70.5 | 28.3 | 14.7 | 53.7 |
| GCD (Vaze et al., 2022) | 51.3 | 48.7 | 56.6 | 39.0 | 29.9 | 57.6 | 35.4 | 27.0 | 51.0 |
| OpenCon (Ours) | **54.7** | **52.1** | **63.8** | **49.1** | **32.7** | **78.6** | **39.3** | **28.6** | **58.9** |

## O  Hardware and Software

We run all experiments with Python 3.7 and PyTorch 1.7.1, using NVIDIA GeForce RTX 2080Ti GPUs.

