# OpenReview forum: "OpenCon: Open-world Contrastive Learning"
_TMLR — Accepted by TMLR_

### Review · Reviewer_6PFD · 2022-10-18

**Summary Of Contributions:**

This work proposes to solve open-world representation learning. In addition to the standard supervised learning loss and the contrastive loss for unsupervised learning, the authors further estimate and update the prototypes of unknown classes to promote the learning of distinguishable features on the unlabeled set.

**Audience:**

Yes

**Broader Impact Concerns:**

In general, I think the contribution and impact of this work are not enough. Although the authors provide clear writing and extensive ablation studies, the core idea is not novel.

**Claims And Evidence:**

Yes

**Requested Changes:**

1. The most change I hope the authors could include is the experiments on the standard ImageNet. Other tasks like detection (as did in Li et al. 2021b) would be good but not required.

2. The authors should also elaborate more on the core contribution and benefit of the proposed method compared with previous EM-like unsupervised learning methods.

**Strengths And Weaknesses:**

Strengths:
1. Paper writing is clear and easy to understand.
2. Ablation study and analysis are comprehensive.

Weakness:
1. I think the proposed method is very similar to the cited Li et al. 2021b, and also shares a similar spirit with previous self-training works [1]. Basically, the idea is to assign labels to high-confident unlabeled data, plus unsupervised learning loss to encourage diverse sample-wise features. I think the concept of prototype learning, even combined with supervised and unsupervised learning, is not novel at this moment.

2. The benchmark is very limited. CIFAR-100 and ImageNet-100 are far from being large enough. For example, most contrastive learning works tested on the standard ImageNet dataset. As this work also involves unsupervised learning, experiments on the standard ImageNet are required for a fair comparison.

3. Lemmas that claims the equivalence between the proposed method and the EM algorithm are not significant. Prototypes are natively centroids in EM. This is also explained in [1].

[1] Domain Adaptation for Semantic Segmentation via Class-Balanced Self-Training.

---

### Review · Reviewer_6dAv · 2022-10-19

**Summary Of Contributions:**

The article proposes an approach, Open-world Contrastive learning (OpenCon) for the task of open-world representation learning. Similarly to semi-supervised learning, in this task, there are both labeled data and unlabeled data but the latter may belong to both known (labeled) and unknown (unlabeled) classes. OpenCon builds on five components:
1. Supervised contrastive loss on the labeled samples.
2. Prototype-based novelty recognition, where all samples very dissimilar (according to a threshold) w.r.t. the existing prototypes are considered as belonging to unknown classes. Note that prototypes for unknown classes are pre-initialized.
3. Contrastive loss on novel classes, where the identified unknown class samples belong to the same positive set if they share the same closest prototype reserved for novel classes.
4. Prototypes update through the moving average of their respective samples.
5. Unsupervised loss (i.e. SimCLR) on unlabeled samples.

Experiments on CIFAR-100 and ImageNet-100 show that OpenCon outperforms the state-of-the-art methods for this setup, ORCA (Cao et al., 2022) and GCD (Vaze et al., 2022).

**Audience:**

Yes

**Broader Impact Concerns:**

The article discusses the broader impact in Appendix A. There are no clear concerns but the ones related to the actual available data: the algorithm may identify unlabeled minority sub-groups if applied to images of humans and/or reflect biases in the data collection (e.g. specific attributes associated only with specific sub-groups). These of course are almost unrealistic setups/cases that would go way beyond the scope for which the algorithm has been developed: they may be nevertheless added to Appendix A for completeness.

**Claims And Evidence:**

Yes

**Requested Changes:**

I am leaning toward recommending the acceptance of the manuscript. Following on the points above, my request is on revising Table 1, and Sections 1-2 (and eventually the title) to better clarify the contribution of the manuscript w.r.t. OCRA and GCD, and avoid that a reader may consider the problem setup as an actual contribution (as correctly not indicated in the last part of Section 1).

As an additional final point: including a discussion of the weaknesses of OpenCon and potential future works in Section 8 would make the article more complete.

**Strengths And Weaknesses:**

**Strengths**:
1. The problem considered in this work (i.e. semi-supervised learning with unlabeled data for unknown classes) is of clear practical relevance due to the usual unconstrained collection of unlabeled data) and interesting since few previous works considered it.
2. The OpenCon framework is tailored to the task (e.g. considering separate loss components for each part of the dataset) and intuitive. It achieves remarkable performance and, due to its simplicity, can serve as a good baseline method for this task.
4. Section 4 provides also a theoretical analysis of OpenCon.

**Weaknesses**:
1. My main concern is that the manuscript currently does not give enough credit to OCRA and GCD, and the fact that the tackled setting is not novel in terms of problem setup. To clarify: the article does not claim that the problem set is new in terms of data but stresses that it is new in terms of learning objective (i.e. representation vs classification accuracy of OCRA). My concern raises from the fact that multiple parts of the manuscript hint at the first statement. Examples are Table 1, where the problem is denoted as "ours", without referring e.g. to the open-world SSL problem pushed by OCRA. The same applies to the second paragraph of Section 1. Similarly for Section 2 both the first sentence and the second paragraph (i.e. difference w.r.t. existing problem setting), something that is clarified only at the end of the section with the explicit reference to OCRA. These statements should be revised (e.g. Sections 1-2, Table 1) and the differences clarified to avoid misunderstandings regarding the contributions.
2. Following on the previous, also GCD has the same objective as OpenCon, i.e. learning good representations for unknown classes (as clarified in Section 5). GCD should also be discussed in Section 2. Eventually, the title could also be made more specific to highlight the main contribution of the work (i.e. OpenCon, the contrastive objective/framework tailored for this problem).
3. In Section 8, it would be interesting to discuss some weaknesses of the model and potential avenues for future work.

**Minors**:
1. I find the analyses in Appendix E (i.e. why not use all unlabeled samples) very insightful, given that how to deal with unlabeled samples is a crucial part of the approach (as shown in Table 4).  Due to its importance, it would be helpful to include part of this analysis in the main manuscript.
2. Typo "in included", the last sentence of Section 8.

---

### Review · Reviewer_wCPe · 2022-10-22

**Summary Of Contributions:**

This paper proposed a new setting, open-world contrastive learning, and a framework accordingly. This setting emphasizes we should utilize both labeled data and unlabeled data from unknown classes for representation learning. A prototype-based framework is proposed with minor modifications to the existing prototype framework and contrastive learning formulation. Some theoretical analyses are included to help understand the proposed framework. They tested in two common benchmarks, CIFAR-100 and IN-100, and shown the proposed method consistently outperforms the baselines.

**Audience:**

Yes

**Broader Impact Concerns:**

There are no concerns about the ethical aspects in the reviewer's mind currently.

**Claims And Evidence:**

Yes

**Requested Changes:**

- Please address all the above concerns, especially the performance in scale-up setting concerns.
- The author claims, "Cao et al. (2022) focus on classification accuracy, but not learning high-quality embeddings.", while they only report classification accuracy. Is there any other evidence to support the proposed framework can learn "stronger" learning representations?
- Table 6, The proposed method under Labeling Ratio = 0.25, $y_l$ = 50 has the same IN-100 Seen results as Labeling Ratio = 0.5, $y_l$ = 50. This is weird, and the reviewer is unsure if there are any inputting errors.
- OCRA achieved higher accuracy in Novel classes under Labeling Ratio = 0.25, $y_l$ = 50 than Labeling Ratio = 0.5, $y_l$ = 50. Could the author please explain the reason or provide analyses here?

**Strengths And Weaknesses:**

The proposed setting/problem/direction is important and more realistic than previous methods. Studying in the OpenWorld with data from unknown classes for which we have no prior knowledge is important. The proposed prototype-based framework is straightforward, and all the components are designed reasonably. Some theoretical analyses are included to provide insights and intuition about the proposed framework. Overall the paper is easy to follow.

Extensive experiments and comparisons are conducted in two datasets, CIFAR-100 and IN-100. An extensible to ViT is included. Proper ablation studies are also conducted to prove the effectiveness of the proposed components.

All current experiments are limited to 100 classes, a huge gap with practical applications. It is not convinced that the proposed method works in this scenery can achieve great performance in scale data.

Since the prototype-based novelty discovery mechanism is one of the key points proposed by this paper, it is probably better to also include ablation studies in IN-100 instead of only testing on CIFAR-100.

For Table 6, it would be better to also include the Labeling Ratio = 0.5, $y_l$ = 50 result for better comparisons.

---

### Review · Reviewer_MY9C · 2022-10-22

**Summary Of Contributions:**

This paper tackles the generalized category discovery problem where the unlabelled data contains classes from both seen classes in the labelled data and novel classes, the goal is to learn to classify all classes.
This paper proposed a contrastive learning frameworks which selects the positive and negative pair via a moving average prototype, good performances are obtained on two datasets.


**Audience:**

Yes

**Claims And Evidence:**

No

**Requested Changes:**

1. In Table 1, the bottom row should be 'Generalized Category Discovery' [R1] or 'Open-World Semi-Supervised Learning' [R2], marking it as 'Ours' does not give enough credit to previous works and may mislead the reader (This problem setting is not new).
2. As shown in [R1], performing generalized category discovery on fine-grained datasets is a more challenging and practical scenario, because the class-wise relation between seen and novel classes are clear (They are all same entry-level classes, e.g. all bird classes for CUB dataset), and the fine-grained differences between classes make it a harder problem to tackle for representation learning, thus I think the paper should include experiments on these datasets to support the claim of state-of-the-art performance.
3. In [R1], the authors also provide strong baselines modified from strong novel category discovery methods, namely RankStat+[R3] and UNO+[R4], it is also shown that UNO+ can outperform GCD in some cases, I think this paper should also include these two baselines for a comparison to give the readers more context. It is also shown in [R1] that a simple k-means on DINO[R8] pretrained features could performs very well, I would suggest the author to include this baseline into the paper.
4. As the overall framework for tackling the GCD problem is very similar to [R5], some more ablations should be conducted in my opinion, such as 1) using the prototypes from k-means instead of moving average, maybe also using the prototypes from a semi-supervised k-means [R1], 2) using the prototypical contrastive learning from [R5] (i.e. pull closer positive prototypes, push away negative prototypes) instead of the instance contrastive learning in the paper, 3) how well does the prototypes from k-means or semi-supervised k-means performs in the prototype-based novelty detection? 4) How does overclustering like in [R5] influences the performance?
5. I think this paper is confusing the use of 'representation learning', in my opinion, 'representation learning' refers to 'learning representations of the data that make it easier to extract useful information when building classifiers or other predictors' [R6], yet this paper only evaluates on classification tasks, it is hard to say if the representation learned is good for transfer learning (which is widely adopted for evaluating representation in computer vision) [R7]. I would suggest the author to tune down the claim about representation learning, and change to a more appropriate title, or try to evaluate the representation learned by the proposed methods using [R7].
6. As [R1] adopts DINO [R8] pretrained checkpoints on ImageNet, and the fact that in practice, one can always have the ImageNet pretrained weights available, I would suggest the paper to have more comparison on the DINO pretrained weights, e.g. apply DINO weights for all methods in table 2.
7. The second sentence in the abstract is not correct in my opinion, the vast majority of representation learning methods indeed could work in the setting this paper is trying to tackle, in fact, the state-of-the-art representation learning methods are self-supervised, thus they do not face the 'open world' problem mentioned in the paper [R9, R10].
8. In my opinion, Lemma 2 is trying to prove that pull positive pairs determined by the moving average prototypes closer is equivalent to aligning the examples to the corresponding prototypes, this is not a new discovery, although novelty is not a criteria for TMLR submission, I would suggest the author to cite previous works already showing this [R11] and [R12].
9. One thing that is not very clear after I read the paper, how is the label assigned to each testing sample? In [R1], the label assign is done by using semi-supervised k-means, in [R2], it is done by a parametric classifier, how does the proposed method do this? Is it k-means or a parametric classifier?

[R1] Generalized Category Discovery, CVPR 2022.

[R2] Open-World Semi-Supervised Learning, ICLR 2022.

[R3] Automatically discovering and learning new visual categories with ranking statistics, TPAMI 2021.

[R4] A Unified Objective for Novel Class Discovery, ICCV 2021.

[R5] Prototypical Contrastive Learning of Unsupervised Representations, ICLR 2021.

[R6] Representation Learning: A Review and New Perspectives, https://arxiv.org/abs/1206.5538

[R7] A Large-scale Study of Representation Learning with the Visual Task Adaptation Benchmark, https://arxiv.org/abs/1910.04867

[R8] Emerging Properties in Self-Supervised Vision Transformers, ICCV 2021.

[R9] Momentum Contrast for Unsupervised Visual Representation Learning, CVPR 2020.

[R10] Masked Autoencoders Are Scalable Vision Learners, CVPR 2022.

[R11] Relax, no need to round: integrality of clustering formulations, ITCS 2015.

[R12] Spectral relaxation for k-means clustering, NeuIPS 2001.

Subjective comments

1. Although this is highly subjective, I would suggest the author to change the last sentence in the abstract, as I think the paper does not provide enough context for the readers, a sentence like this may mislead the readers.
2. I would suggest the author to tune down the claims made in the abstract, as some of them may not be correct as already mentioned previously.



Formatting issues

1. The first equation in sec 4.1 should remove the argmax, this equation is comparing the value of the log likelihood, not the argmax of the log likelihood. The same goes for the equation in Lemma 4.2
2. Last sentence of sec 8, 'in included' -> 'is included'


**Strengths And Weaknesses:**

S1: The proposed framework seems to be performing well.

S2: Other than the generalized category discovery problem, the proposed framework also brings good performance for OOD detection, which is very interesting.


W1: The paper itself has many overclaims, which may mislead the readers and does not give enough credit for existing works.

W2: Some important baselines are missing, see Requested changes.

W3: Also some important datasets for evaluating generalized category discovery are missing.

---

### Decision · Action_Editors · 2022-11-27

**Recommendation:** Accept as is

**Comment:**

The paper provides an algorithm for performing representation learning in an open world semi-supervised setting (i.e. with both known and unknown classes). The paper provides good empirical evidence for the efficacy of the approach on a number of tasks. The paper also links OpenCon to the EM algorithm to provide further justification for the approach.

Initially, reviewers had concerns with the positioning of the paper with respect to prior work on this problem, but in the updated version the claims have been revised to address these concerns. The reviewers also raised concerns with narrow experiments. With the additional of a larger dataset (full ImageNet), fine-grained recognition problems, additional baselines, and additional ablations, these have been adequately addressed.

Open world SSL / Generalized Category Discovery is a topic of interest to the community, and this paper makes a good contribution to the area.

Congratulations on the acceptance!

**Audience:**

Yes. This paper pertains to multiple topics of substantial interest to the community at the moment: representation learning, semi-supervised learning, novel, and out-of-distribution detection. The paper positions itself appropriately and compares to prior contributions, has impressive-looking performance, and a quite extensive analysis, therefore, I believe will be of interest to some of TMLR's audience.

**Claims And Evidence:**

The paper claims to introduce a new algorithm (OpenCon) that performs well for representation learning with semi-supervised data in an open-world setting. This claim appears well-substantiated. The paper provides a theoretical link between OpenCon and the EM algorithm. It also provides ample experimental evidence to support the efficacy of the OpenCon: Some medium-scale experiments (C-100, ImageNet-100) comparing to previous approaches in a standardized setup, larger-scale ImageNet experiments, fine-grained recognition, and additional ablations/analysis.

At first, the reviewers had concerns that sufficient credit was not given to prior work (e.g. Generalized Category Discovery) that addressed a similar problem, and that the experiments were too narrow. These concerns have been adequately addressed with revised framing of the contributions and additional experiments.

---

> ### Author Response · Authors · 2022-11-29
> **thank you**
>
> Dear Editor,
>
> We thank you for the thorough notes summarizing the review process, and for accepting our manuscript. We are really glad the work is appreciated by you and the reviewers.
>
> We will do a final polish and submit the camera ready accordingly.
>
> Sincerely,
> Authors